# EXPLAINING POINT PROCESSES BY LEARNING INTERPRETABLE TEMPORAL LOGIC RULES

**Shuang Li**[1] [*][†] **, Mingquan Feng**[2] [*][‡] **, Lu Wang**[3] **, Abdelmajid Essofi**[4] **,**
**Yufeng Cao**[2] [§] **, Junchi Yan**[2] [‡] **, Le Song**[4,5]
[1]CUHK, Shenzhen    [2]Shanghai Jiao Tong University    [3]Microsoft Research    [4]MBZUAI    [5]BioMap
lishuang@cuhk.edu.cn, {fengmingquan, yufeng.cao, yanjunchi}@sjtu.edu.cn,
wlu@microsoft.com, {Abdelmajid.Essofi, le.song}@mbzuai.ac.ae

## ABSTRACT

We propose a principled method to learn a set of human-readable logic rules to explain temporal point processes. We assume that the generative mechanisms underlying the temporal point processes are governed by a set of first-order temporal logic rules, as a compact representation of domain knowledge. Our method formulates the rule discovery process from noisy event data as a maximum likelihood problem, and designs an efficient and tractable branch-and-price algorithm to progressively search for new rules and expand existing rules. The proposed algorithm alternates between the rule generation stage and the rule evaluation stage, and uncovers the most important collection of logic rules within a fixed time limit for both synthetic and real event data. In a real healthcare application, we also had human experts (i.e., doctors) verify the learned temporal logic rules and provide further improvements. These expert-revised interpretable rules lead to a point process model which outperforms previous state-of-the-arts for symptom prediction, both in their occurrence times and types. [1]

## 1 INTRODUCTION

Event sequences with irregular time intervals are ubiquitous. The inter-event times usually convey rich information regarding the underlying dynamics such as disease progression (Liu et al., 2015). It is useful to understand events' generating mechanisms, as well as the occurrence reason and time. In systems where domain knowledge is rich, events generating can usually be governed by a few *first-order temporal logic rules*, as a compact representation of knowledge. In healthcare, the knowledge "if a sudden fall in blood pressure is observed, vasopressors are required to be applied to patients immediately; and then the blood pressure may return to normal afterwards" may explain why the event "blood pressure returns to normal from the abnormally low level " is observed after the event "vasopressors are used". This domain knowledge can be summarized as a collection of logic rules with *temporal relation constraints*, such as "A happens *before* B", "If A happens, and *after* 5 mins, B can happen", and "If A and B happen *simultaneously*, then *at the same time* C can happen". However, temporal logic alone is not an ideal temporal model, as hard constraints will be too strict to model the recurrent noisy event data.

Meanwhile, a large amount of literature has been devoted to modeling event data, among which *temporal point process* (TPP) models provide an elegant framework without the need to discretize the time horizon into bins and to compute the count of events within each bin. TPP models treat the inter-event time as random variables and directly model the intensity function (i.e., occurrence rate) of the events. However, most TPP models lack interpretability, and they can not represent domain knowledge in a human-readable form. Recently, Li et al. (2020) proposed a unified framework to marry *point*

---

[*]Equal contribution.

[†]Shuang Li is with the School of Data Science, The Chinese University of Hong Kong, Shenzhen, Shenzhen 518172, China, and the Shenzhen Institute of Artificial Intelligence and Robotics for Society, Shenzhen 518129, China (corresponding author, e-mail: lishuang@cuhk.edu.cn).

[‡]Mingquan Feng and Junchi Yan are with Dept. of CSE, MoE Key Lab of Artificial Intelligence, and SJTU-Yale Joint Center for Biostatistics and Data Science, National Center for Translational Medicine, SJTU.

[§]Yufeng Cao is with the Antai College of Economics and Management and the Data-Driven Management Decision Making Lab, SJTU.

[1]Code is available at https://github.com/FengMingquan-sjtu/Logic_Point_Processes_ICLR

*process intensity functions* with *temporal logic rules*. This method employs a set of pre-specified temporal logic rules to design the intensity functions to incorporate domain knowledge. The resulting so-called *temporal logic point process* models are inherently interpretable and expressive. Using such logic-informed intensity functions, the model can capture nonlinear dependencies, interactions, and various temporal relations between events. It was also shown that, by specifying one or two simple temporal logic rules, many existing parametric models, e.g. the Hawkes process (i.e. self-exciting processes) (Hawkes, 1971a) and the self-correcting process (Isham & Westcott, 1979) are special cases of the proposed temporal logic point process.

However, in (Li et al., 2020) the temporal logic rules are required to be specified by the model builder, but they may not be known beforehand. Models constructed in such a way may suffer from model misspecification when the pre-specified logic rules are incorrect or missing.

*Can we automatically discover the temporal logic rules governing the event dynamics based on historical event data alone?* It is challenging as the space of possible temporal logic formulas is huge, comprising of combinations of massive discrete logic variables and all kinds of temporal relations.

Previous inductive logic programming approaches focused mostly on deriving the relations between discrete logic variables, but have largely ignored the crucial temporal information (Dash et al., 2018; Wei et al., 2019). A generalization of these methods by further considering temporal relations in logic variables is largely missing and is in pressing needs due to the increasing availability of event data. To address the aforementioned challenges, we propose the TEmporal Logic rule LearnER (TELLER) algorithm for learning temporal logic rules from event sequences. TELLER is inspired by the branch-and-price algorithm (Barnhart et al., 1998), a column generation (CG) algorithm for linear programming (LP) problems where the number of variables is too large to be considered explicitly. In the CG algorithm framework, the original LP problem is solved via two alternating procedures: the master problem and the subproblem, where the master problem is the original problem with only a subset of variables being considered and the subproblem is a new problem created to identify a new variable to be added. Similarly, TELLER also alternates between solving a master problem and a subproblem, where the master problem aims to evaluate the current rules by maximizing the likelihood and reweighting these rules as in (Li et al., 2020), and the subproblem is to search and construct a new temporal logic rule (by extending a current rule or adding a new short rule). It repeats until the likelihood can no longer be improved by adding new candidate rules. In this way, TELLER searches through the vast space of potential temporal logic rules and learns the importance weights of the discovered rules to hedge against noise in the data. Fig. 1 shows the flow.

Specifically, TELLER is computationally efficient and with guarantees. It includes the candidate rules one by one in a tractable and progressive fashion, and is able to find a near-optimal set of rules. The hypothesized logic rules are in disjunctive normal form (DNF, OR-of-ANDs) with temporal relation constraints, and can be of various lengths. Given the temporal logic point process modeling framework, we show that the resulting objective function, i.e. likelihood, is convex, which guarantees optimal performance for our search algorithm. A set of logic rules and their importance weights are jointly learned by maximizing the likelihood. The most important collection of logic rules are guaranteed to be discovered within a fixed time limit.

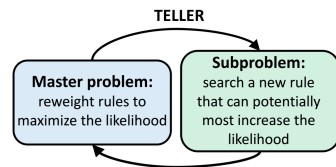

Figure 1: The flow of TELLER. It alternates between a rule evaluation stage (master problem) and a rule proposal stage (subproblem).

The uncovered rules by TELLER will not only shed light on *when* and *what* events would happen, but also *why* some events would happen at a specific time. Our method facilitates the human-readable knowledge exchange between experts and point process models. On one hand, the mined temporal logic rules may supplement or refine the existing knowledge; on the other hand, human experts can easily provide feedback to modify the learned models via logic rules, as a way to add the safety of using these models in high-stake tasks like healthcare and autonomous driving. In our real experiment related to medicine, TELLER is used to learn the explanatory temporal logic rules regarding the choice and arrangement of drugs in improving patients' health status. Doctors were asked to justify the rules and they confirmed that these discovered logic rules are consistent with the pathogenesis and have captured the most important factors in affecting patients' health status.

## 2 BACKGROUND

### 2.1 FIRST-ORDER TEMPORAL LOGIC RULES

First-order temporal logic is a form of symbolized reasoning in which each statement is a composition of temporal predicates and their relations. We formally define the interval-based temporal logic below.

This type of temporal logic rule is often suitable for reasoning about events with duration, which are better modeled if the underlying temporal ontology uses time intervals. It fits well with the temporal point process models for event sequences, where the time intervals of events are explicitly modeled. Refer to (Goranko & Rumberg, 2021) for a comprehensive survey of various temporal logic models.

**Temporal predicate.** First define a set of *static predicates* $\mathcal{X} = \{X_1, \ldots, X_d\}$, where each static predicate $X_i(c)$ is a logic random variable that defines the *property* or *relation* of entities, such as $\mathrm{Smokes}(c)$ or $\mathrm{Friend}(c, c')$, where $c$ and $c'$ are the entities. By adding a temporal dimension to predicate $X_i$, we obtain a *temporal predicate* $\{X_i(c, t)\}_{t \geq 0}$, which can be viewed as a continuous-time stochastic process. Given observations, each temporal predicate will be grounded as a list of ordered 0-1 events, which take True (1) and False (0) in an alternating way with the state transition times recorded. For example, a temporal predicate $\{\mathrm{NormalBloodPressure}(c, t)\}_{t \geq 0}$, where entity $c$ is referring to the patient, will take value 1 or 0 at any time $t$ to indicate whether the patient's blood pressure is normal or not, with stochastic state transition time (see Fig. 2).

To simplify the notation, we will temporally drop the dependency of predicates on entities. We will write $X(c, t)$ as $X(t)$ instead and the grounded temporal predicate as $\{x(t)\}_{t \geq 0}$.

## 2.2 TEMPORAL LOGIC POINT PROCESS

Temporal Point Process (TPP) provides an elegant tool to capture the dynamics of the event sequences. It is characterized by conditional intensity function, denoted by $\lambda(t | \mathcal{H}_t)$, where history $\mathcal{H}_t$ is the knowledge of times of all events. By definition, we have $\lambda(t | \mathcal{H}_t) dt = \mathbb{E}[N([t, t+dt]) | \mathcal{H}_t]$, where $N([t, t+dt])$ denotes the number of points falling in an interval $[t, t+dt]$. Here we aim to use TPP to model these 0-1 events and use the intensity function to capture the 0-1 transition rate.

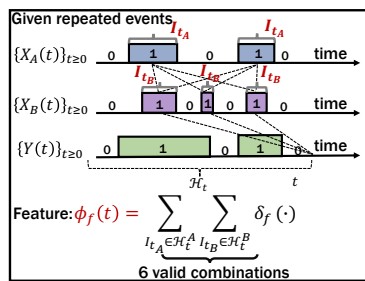

Figure 2: Feature constructions of TLPP using a simple logic formula $f: Y \leftarrow A \wedge B$, as a template to gather combinations of the body predicate history events. $A$ has 2 events and $B$ has 3 events, leading to 6 valid combinations.

**Temporal relation.** We will use the *interval-based temporal relations* originally introduced in Allen's seminal paper (Allen, 1990) to describe the temporal relations that can exist in any two temporal predicates. There are 13 types of possible temporal relations, including *Before, Meets, Overlap* and so on (see Appendix B for a comprehensive illustration), and we denote this set by $\mathcal{R}$. To evaluate these temporal relations, we need to define the *time interval* using the state transit in/out times. For example, at time $t_A$, the temporal predicate $\{X_A(t)\}_{t \geq 0}$ takes state $x_A = 0$ or 1, we define $I_{t_A} = (t_{A_1}, t_{A_2}]$, $t_A \in (t_{A_1}, t_{A_2}]$, as a time interval where $t_{A_1}$ is the transition time that this predicate enters the state $x_A$ and $t_{A_2}$ is the transition time that this predicate leaves the state $x_A$. For any two temporal predicates, given their time intervals, $I_{t_A} = (t_{A_1}, t_{A_2}]$ and $I_{t_B} = (t_{B_1}, t_{B_2}]$, their temporal relation function, denoted as $R_{A \, type \, B}(I_{t_A}, I_{t_B}) \in \mathcal{R}$, is a logic function defined over the time intervals and can be evaluated by plugging in the specific time intervals. For example, $R_{A \, Before \, B} = \mathbb{1}\{t_{B_1} > t_{A_2}\}$ and $R_{A \, Equals \, B} = \mathbb{1}\{t_{A_1} = t_{B_1}\}\mathbb{1}\{t_{A_2} = t_{B_2}\}$. All these definitions of temporal relations are also given in Appendix B.

**Temporal logic formula.** A first-order temporal logic rule $f$ is defined as logical connectives of *temporal predicates* and their *temporal relations*. The generic form is:

$$f: (\neg) Y(t_y) \leftarrow \bigwedge_{X_u \in \mathcal{X}_f^+} X_u(t_u) \bigwedge_{X_v \in \mathcal{X}_f^-} \neg X_v(t_v) \bigwedge_{X_u, X_v \in \mathcal{X}_f} R_{u \, type \, v}(I_{t_u}, I_{t_v}) \quad (1)$$

where each temporal predicate and the temporal relations can be evaluated at $t_y$, $t_u$ and $t_v$ along with their corresponding intervals $I_{t_u}$ and $I_{t_v}$. In the above formula, negation ($\neg$) means the head predicate $Y$ can take a negative sign; $\mathcal{X}_f = \mathcal{X}_f^+ \cup \mathcal{X}_f^-$ is the set of predicates defined in $f$, where $\mathcal{X}_f^-$ is the set of predicates as negation in the formula $f$, and $\mathcal{X}_f^+ = \mathcal{X}_f \setminus \mathcal{X}_f^-$. We assume $\leftarrow$ has a causal direction, and the body part of the formula indicate the evidence to be gathered from history to deduce the state of the head predicate. By this assumption, it is only valid to consider $t_y \geq t_u, t_v$.

Given observed sequences of data, each predicate in $\mathcal{X}$ will be grounded as a list of ordered 0-1 events. The main idea of *temporal logic point process* (TLPP) (Li et al., 2020) is to use the temporal logic rules (1) as templates to selectively choose *combinations of events* from history as evidence to infer the transition rate of these 0-1 events. In this way, the structure of the intensity function will be informed by the pre-specified temporal logic rules. Below we provide more details.

**Logic-informed intensity function.** Let $\mathcal{H}_t$ include all the historical trajectories of the grounded predicates in $\mathcal{X}$ up to $t$. TLPP leverages the rule set to model the state transition dynamics of the head predicates. Define $\lambda^*(t) := \lambda(t | \mathcal{H}_t)$ as the transition intensity for $\{Y(t)\}_{t \geq 0}$ to transit from 0 to

1, given history up to $t$; and $\mu^*(t) := \mu(t|\mathcal{H}_t)$ vice versa. Assume the generating process of the 0-1 events are governed by the set of temporal logic rules $\mathcal{F} = \{f_1, f_2, \dots\}$.

First consider one formula $f$, to incorporate its knowledge in intensity construction, we define a *formula effect* (FE), which aims to gather only the effective combinations of the historical predicate events defined by $f$ as evidence to reason about the transition rate of the head predicate. To ease the notation, define the predicate index set of $\mathcal{X}_f$ as $U$, then the *formula effect* of $f$ is computed as

$$\delta_f\left(t \mid y(t), \{x_u(t_u), I_{t_u}\}_{u \in U} \in \mathcal{H}_t\right) := f\left(1 - y(t), \{x_u(t_u), I_{t_u}\}_{u \in U}\right) - f\left(y(t), \{x_u(t_u), I_{t_u}\}_{u \in U}\right)$$
(2)

where $y(t)$ is the observed head predicate state at $t$, $1 - y(t)$ is its counterfactual state, $\{x_u(t_u), I_{t_u}\}_{u \in U} \in \mathcal{H}_t$ is one historical combination of the body predicates (including their states and time intervals) in $f$, and $f(\cdot)$ is the clausal form of (1), which is an alternative expression to ease the evaluation of the rule. For example, $Y \leftarrow X$ is logically equivalent to its clausal form $\neg Y \vee X$. FE, which is essentially a difference between the what-if scenario and the true scenario, answers the question such as "should $Y(t)$ transit its state given logic formula $f$ is true".

One can check that the sign of FE can only be 1, -1 or 0, which can be interpreted as: $\text{sign}(\text{FE}) = 1$ indicates a positive effect for the head to transit, -1 indicates a negative effect, and 0 means no effect. Only the non-zero FE will refer to an effective combination. Let $\mathcal{H}_t^u$ be the historical trajectory specific to predicate $u$ up to $t$, and one can aggregate all the valid (i.e., non-zero) FEs from history by a summation over all combinations of the temporal predicate states and their intervals, i.e.,

$$\phi_f(t) = \sum\nolimits_{\{(x_u(t_u), I_{t_u}) \in \mathcal{H}_t^u\}_{u \in U}} \delta_f(t \mid y(t), \{x_u(t_u), I_{t_u}\}_{u \in U}),$$
(3)

where $\phi_f(t)$ is the *feature* informed by logic formula $f$. One can refer to Fig. 2 for an illustration.

For each $f \in \mathcal{F}$, one can build the rule-informed and history-dependent features $\phi_f(t)$ as above, and assumes the rules are connected in disjunctive normal form (OR-of-ANDs) to deduce $Y$. Then the transition of $\{Y(t)\}_{t \geq 0}$ are modeled as monotonically increasing and non-negative functions of the weighted sum of the features:

$$\lambda^*(t) = \exp\left(b_0 + \sum\nolimits_{f \in \mathcal{F}} w_f \cdot \phi_f(t)\right), \qquad \mu^*(t) = \exp\left(b_1 + \sum\nolimits_{f \in \mathcal{F}} w_f \cdot \phi_f(t)\right)$$
(4)

where $\boldsymbol{w} = [w_f]_{f \in \mathcal{F}} \geq 0$ are the weight parameters associated with each rule $f \in \mathcal{F}$, and $b_0$, $b_1$ are the spontaneous intensity, which are distinct for the dual intensities. Weight parameters will be shared by the dual intensities, but the calculated features $\phi_f$ through (2) and (3) will always have opposite signs. One can think of the formula weight $w_f$ as the confidence level put on $f$. The higher the weight, the more influence that the formula will have on the intensity. The rationale of TLPP is that the intensity functions are constructed in a way that the yielding 0-1 event sequences will enable the set of logic rules to be more satisfied.

## 3 THE PROPOSED METHOD: TELLER

**Likelihood function and its convexity.** Given $\{\mathcal{H}_t\}_{0 < t < T}$, one can write out the likelihood w.r.t. the intensity $\lambda^*(t)$ and $\mu^*(t)$. Suppose a realization of $\{Y(t)\}_{0 < t < T} = \{Y(0) = 0, Y(t_1) = 1, \dots, Y(t_n) = 1, t_n < T < t_{n+1}\}$, the (log) likelihood are (see proof in Appendix C): $\ell(\boldsymbol{w}, b_0, b_1) = \log \mathcal{L}(\boldsymbol{w}, b_0, b_1)$, where:

$$\mathcal{L}(\boldsymbol{w}, b_0, b_1) = \lambda^*(t_1) \exp\left(-\int_0^{t_1} \lambda^*(s) ds\right) \cdot \mu^*(t_2)$$
(5)
$$\exp\left(-\int_{t_1}^{t_2} \mu^*(s) ds\right) \cdots \exp\left(-\int_{t_n}^{T} \mu^*(s) ds\right),$$

Note that $-\ell$ is convex w.r.t $\boldsymbol{w}$, $b_0$ and $b_1$. This is true as the intensity function has a functional form of Eq. (4), and this turns the log-likelihood function into a generalized linear model (GLM) with Poisson observations and log link (Fahrmeir & Tutz, 2013). It is well-known that the negative log-likelihood of a GLM is convex w.r.t the model parameters. This convexity property leads to a convergence guarantee for TELLER.

TELLER uses TLPP as the backbone to evaluate the likelihood of the grounded 0-1 temporal predicate sequences. The likelihood will be maximized to jointly learn the set of temporal logic formulas and their weights.

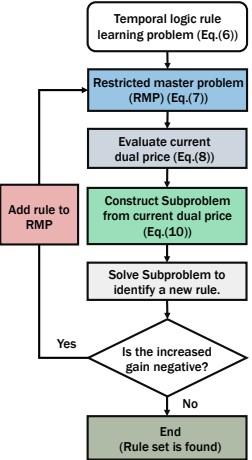

Figure 3: TELLER

The problem is essentially combinatorial and requires enumerating an exponentially large set of combinations of the predicates and their signs. Our problem is even more challenging compared to traditional inductive logic programming due to the sequential properties of data with temporal information. TELLER solves this problem using the procedures as shown in Fig. 3. We will start our exposition with the formulation of the original, or the master rule learning problem.

## 3.1 TEMPORAL LOGIC RULE LEARNING PROBLEM (MASTER PROBLEM)

We aim to uncover the set of temporal logic rules $\mathcal{F}$ based via optimizing:

$$P_{\text{Master}}: \quad \boldsymbol{w}^*, b_0^*, b_1^* = \underset{\boldsymbol{w}, b_0, b_1}{\operatorname{argmin}} -\ell(\boldsymbol{w}, b_0, b_1) + \lambda_0 \sum_{f \in \bar{\mathcal{F}}} c_f w_f; \quad s.t. \quad w_f \geq 0, \quad f \in \bar{\mathcal{F}} \quad (6)$$

where $\bar{\mathcal{F}}$ is the complete and exponentially large collection of *all possible* temporal logic formulas that can be created by the set of the pre-specified temporal predicates and their temporal relations. The algorithm needs to determine: 1) the assignment $\mathcal{X}_f$, $\mathcal{X}_f^+$, and $\mathcal{X}_f^-$; 2) the important temporal relations defined over the involved predicates in $f$; and 3) the weight parameter $w_f$ as in (4). The regularization penalties related to the rule complexity are incorporated into the objective to force sparsity and to trade off accuracy against rule simplicity. Rule complexity coefficient $c_f$ can be the rule length if we want to learn both the sparse and short rules. Any affine function e.g. $\lambda_0 \sum_{f \in \bar{\mathcal{F}}} c_f w_f$ with $c_f \geq 0$, $\lambda_0 \geq 0$ should work here and will not change the convexity. One can further tune $\lambda_0$ to balance the negative likelihood and the complexity penalty via cross-validation. The sparsity of $\boldsymbol{w}^*$ will indicate the collection of rules learned and these non-zero weight rules will compose $\mathcal{F}$.

**Key Idea:** Although we can write the rule learning problem as a regularized convex optimization above, the set of variables $(w_f)$ is exponentially large and can not be optimized simultaneously in a tractable way. The idea of TELLER is to start from a restricted master problem (RMP), where the search space is smaller and the solution is tractable but suboptimal. TELLER will gradually improve this solution by iteratively expanding the search space until the solution approximates the optimum. The idea is inspired by the fact that most of the candidate rules in $\bar{\mathcal{F}}$ will not be in $\mathcal{F}$ and will lead to a zero value weight in the optimal solution. We do not want to waste resources by including these redundant rules when we expand the search space to optimize (6). TELLER will always generate the rule that has the potential to improve the objective function, which is an any-time algorithm with the guarantee that the most important rules have been considered.

Our progressive algorithm gives rise to two questions: **1)** the criterion to determine the new rule to add; **2)** How to construct new candidate temporal logic rules, as will be tackled in the following.

## 3.2 CRITERION TO ADD RULES

**Restricted master problem (RMP).** TELLER first replaces the original rule set $\bar{\mathcal{F}}$ by a small subset $\mathcal{F}_0 \subset \bar{\mathcal{F}}$ and gradually expands this subset to produce a nested sequence of subsets $\mathcal{F}_0 \subset \mathcal{F}_1 \subset \cdots \subset \mathcal{F}_k \subset \cdots$. It does so by adding candidate rules identified by subproblems (we will elaborate on this later). Note that the initial rule set $\mathcal{F}_0$ can be an empty set or any pre-defined small set.

For each $\mathcal{F}_k$, $k = 0, 1, \ldots$, TELLER solves the restricted master problem by replacing $\bar{\mathcal{F}}$ with $\mathcal{F}_k$:

$$P_{\text{Restricted}}: \quad \boldsymbol{w}^*_{(k)}, b^*_{0,(k)}, b^*_{1,(k)} = \underset{\boldsymbol{w}, b_0, b_1}{\operatorname{argmin}} -\ell(\boldsymbol{w}, b_0, b_1) + \sum_{f \in \mathcal{F}_k} c_f w_f; \quad s.t. \; w_f \geq 0, \; f \in \mathcal{F}_k. \quad (7)$$

This can be regarded as the *rule evaluation stage*, where all rules in the current set will be reweighed. Note that an *optimal* solution to the *restricted* master problem above can be extended to a *feasible* solution to the original problem by setting the weights of all missing $f \in \bar{\mathcal{F}} \setminus \mathcal{F}_k$ to zero. The *optimality* of this extended solution can be verified by leveraging the convexity property of the original problem (6). Let the Lagrangian of the original problem be $L(\boldsymbol{w}, b_0, b_1, \boldsymbol{\nu}) = -\ell(\boldsymbol{w}, b_0, b_1) + \sum_{f \in \bar{\mathcal{F}}} c_f w_f - \sum_{f \in \bar{\mathcal{F}}} \nu_f w_f$, where $\nu_f \geq 0$ is the Lagrange multiplier associated with the non-negativity constraints of $w_f$. Being a convex problem, strong duality holds under mild conditions. Suppose $\boldsymbol{w}^*, b_0^*, b_1^*$ is primal optimal, and $\boldsymbol{\nu}^*$ is dual optimal, then $-\ell(\boldsymbol{w}^*, b_0^*, b_1^*) = \inf_{\boldsymbol{w}, b_0, b_1} L(\boldsymbol{w}, b_0, b_1, \boldsymbol{\nu}^*)$. This implies the *complementary slackness*, which will lead to the following condition.

**Evaluate current dual price.** Obtain the dual price for each constraint in the *original problem (6)*:

$$\nu_{f,(k)} = -\left.\frac{\partial \ell(\boldsymbol{w}, b_0, b_1)}{\partial w_f}\right|_{\boldsymbol{w}^*_{(k)}, b^*_{0,(k)}, b^*_{1,(k)}} + c_f, \quad (8)$$

where subscript $(k)$ means the price depends on the current optimal solution of the RMP, and subscript $f$ refers to the constraint $w_f \geq 0$. The optimality condition says: if the extended solution of $\boldsymbol{w}^*_{(k)}$ is optimal to the original problem, it must satisfy the following condition:

$$w^*_{f,(k)} > 0 \; \Rightarrow \; \nu_{f,(k)} = 0; \qquad w^*_{f,(k)} = 0 \; \Rightarrow \; \nu_{f,(k)} \geq 0 \qquad \forall f \in \bar{\mathcal{F}}, \quad (9)$$

where $\nu_{f,(k)}$ is computed via (8). For the missing rules, we have $w_{f,(k)} = 0$ automatically, and if they are the optimal solution to the original problem, the computed $\nu_{f,(k)} \geq 0$ must hold. A more detailed description of complementary slackness and optimality condition is given in Appendix D.

Therefore, the rule adding criterion is to search over the missing rules and find the rule that most *violates* the condition, i.e., leads to the most negative $\nu_f$ by (8). Adding it to $\mathcal{F}_{k+1}$ will most reduce the objective value. Assuming the computed minimal $\nu_f \geq 0$, it is guaranteed that the current solution to the restricted problem is also optimal to the original one. A subproblem is constructed and optimized to identify a new rule, as will be discussed next.

### 3.3 Propose a New Temporal Logic Rule

**Construct a subproblem from the current dual price.** A subproblem is formulated to propose a new (or missing) temporal logic rule which can potentially improve the optimal value of the RMP, i.e. having a negative increased gain. The increased gain of a missing rule is defined as the possible change in the objective per unit when it is included in the rule set, and it is computed by taking the partial derivative of the objective function w.r.t the rule weight. Note the subproblem itself is a minimization, to find the most negative increased gain.

Given the solution $\boldsymbol{w}^*_{(k)}, b^*_{0,(k)}, b^*_{1,(k)}$ for the restricted master problem (7), a subproblem is constructed by taking the partial derivative w.r.t the weights. The log likelihood can be regarded as a function of the dual conditional intensities: $\ell(\lambda^*, \mu^*)$. By the chain rule, the subproblem optimizes:

$$\min_{\phi_f \in \Phi} \nu(\phi_f | \boldsymbol{w}^*_{(k)}, b^*_{0,(k)}, b^*_{1,(k)}) = -\left( \frac{\partial \ell(\lambda^*, \mu^*)}{\partial \lambda^*} \frac{\partial \lambda^*}{\partial w_f} + \frac{\partial \ell(\lambda^*, \mu^*)}{\partial \mu^*} \frac{\partial \mu^*}{\partial w_f} \right)\Bigg|_{\boldsymbol{w}^*_{(k)}, b^*_{0,(k)}, b^*_{1,(k)}} + c_f, \quad (10)$$

where $\phi_f \in \Phi$ is the rule-based feature, $\lambda^*$ and $\mu^*$ are functions of $\phi_f$, and the conditional intensities are time and history-dependent (see Eq. (4)). In our formulation, this subproblem objective happens to be the current dual price. The objective can be hard to express in a closed-form but will be easily evaluated numerically. Minimizing over the feature space $\Phi$ is essentially searching through a combinatorial space of the temporal predicates and their temporal relations to determine a rule $f$, so that its yielding feature will minimize the objective function.

**Solve subproblem to identify a new rule.** As discussed above, if the optimal value of the subproblem is negative, we have identified a feature (i.e., rule) to enter the rule set to construct $\mathcal{F}_{k+1}$; if the optimal value of the above problem is non-negative, we have proven that the current solution to the RMP is also optimal to the original problem. Note that the subproblem would not generate the same feature or rule more than once, since the optimal Lagrangian multipliers in the RMP are non-negative. However, explicitly solving subproblem (10) requires enumerating all possible conjunctions of the input predicates, their signs, and all possible pairwise temporal relations. The construction of feature $\phi_f$ requires evaluating all valid combinations of predicate events from history.

**Speedup subproblem.** To make the subproblem tractable, we adopt a couple of principles. 1) Theoretically, the *optimality of the subproblem can be sacrificed*. In fact, any solution to (10) that has *a negative objective* will generate a rule with negative increased gain and can be added to the rule set. This provides a performance guarantee for speedup. 2) Practically, *prior knowledge in rule structures can be leveraged*. Our prior knowledge has 3 parts: *sparsity*, *heredity*, and *expert's preference* in rule templates. *sparsity* means the uncovered rule set is small and the rule lengths are generally short. In our algorithm, the maximal rule length is pre-specified as $H$. It means we do not need to consider rules with lengths exceeding $H$. By *heredity*, we mean if the conjunction of the input predicates is important, at least one of the involved input predicates will show some significance. Heredity justifies that we can gradually grow rules from the existing short segments in the current set.

**Search algorithms.** With the prior knowledge above, we can design search algorithms in a way like depth-first search (DFS) or breadth-first search (BFS). In our setting, they are named as rule-extension-first-search (REFS) and rule-addition-first-search (RAFS), respectively. REFS will always extend the existing rules until the length reaches $H$, while RAFS will first add all possible rules with length 1, and then consider the rules with length 2 and so on. Here we describe details of REFS, and leave RAFS for Appendix F. In Fig. 4, a REFS is used to construct a new temporal logic rule. Specifically, it starts by constructing a rule with length one (i.e., one body predicate). This is achieved by scanning the predicate set $\mathcal{X}$ and temporal relation set $\mathcal{R}$, then enumerating the signs and possible temporal relations with the head predicate. The subproblem objective is used to score these segments and select the most negative one. The process continues to extend current segments in the same way, until the rule length reaches $H$, or the minimal subproblem objective becomes positive. In this way, the complexity of our subproblem is $\mathcal{O}(d)$, where $d$ is the number of predicates. Once new rules are added to the rule set, the model will be re-fitted and the model parameters will be re-weighted according to Eq. (7). Longer rules might outweigh their short segments after the rule evaluation stage and thus the length of an important rule can be determined.

In addition, we can use *expert's preference* in rule templates to solve the subproblems. Human experts provide preferred rule templates (rather than a rule set), which reduce the rule searching space, since it masks out many irrelevant predicates and their combinations. For example, doctors are interested in how a treatment affects the evolution of patient health status. To reflect the doctor's conjectures, the rule template can be defined as "symptoms ← drugs ∧ symptoms".

### 3.4 DISCUSSION ON SCALABILITY OF THE ALGORITHM

The previous discussion provides a theoretical framework to find an (near-) optimal temporal logic rule set based on the point process model. For large problems with thousands of event sequences and hundreds of predicates, this type of progressive algorithm tends to be hard to terminate, and obtaining the optimality condition is almost impossible; this is the so-called tailing-off effect (Lübbecke & Desrosiers, 2005; Savelsbergh, 2002). Therefore, we need other techniques to make the TELLER scale with the data and the predicates. For example, we may condition an early termination to the algorithm if a fixed time limit is exceeded. In practice, setting the time limit is proven to work well and the most important rules are likely to be discovered. For large problems, we may also choose to take a random subset of sequences and a random

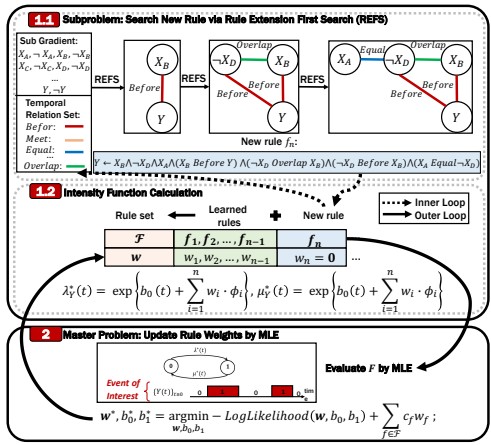

Figure 4: Overall flow of TELLER. It alternates between rule evaluation stage and rule proposal stage. The logic rules and their weights are jointly learned.

subset of predicates for evaluation. When solving for the RMP, we can use a stochastic gradient descent-type of optimization to approximate the gradient using a batch of sequences. By doing so, we enable the RMP to scale with the data (sequences). Furthermore, we can warm-start the model parameters for consecutive RMPs, i.e. we initialize $w_{(k+1)}, b_{0,(k+1)}, b_{1,(k+1)}$ using the optimal solution $w^*_{(k)}, b^*_{0,(k)}, b^*_{1,(k)}$. When solving the subproblems, besides the efficient search scheme, we may also choose to take a subset of the predicates in searching for further speedup.

When the domain knowledge is not accessible, a rule of thumb to initialize $\mathcal{F}_0$ is to start with an "empty" set, which refers to there being only a base term in the intensity function, i.e., $\lambda^*(t) = \exp(b_0)$ and the initial estimate can be learned by MLE (i.e., restricted master problem).

## 4 EXPERIMENTS

We evaluate TELLER on synthetic and real event data. For real data, we consider healthcare treatment understanding and crime pattern learning (crime results are in Appendix H & I).

### 4.1 BASELINES

We considered the following SOTA baselines: 1) Recurrent Marked Temporal Point Processes (RMTPP) (Du et al., 2016b), the first neural point process (NPP) model, where the intensity function is modeled by a Recurrent Neural Network (RNN); 2) Neural Hawkes Process (NHP) (Mei & Eisner, 2016), an improved variant of RMTPP by constructing a continuous-time LSTM; 3) Transformer Hawkes Process (THP) (Zuo et al., 2020), an NPP model with a self-attention mechanism; 4) Tree-Regularized GRU (TR-GRU) (Wu et al., 2018a), a deep time-series model with a designed tree regularizer to add model interpretability; 5) Recurrent Point Process Network (RPPN) (Xiao et al., 2019), where the intensity is modeled by two interleaved RNNs with attention mechanism; 6) CAUSE (Zhang et al., 2020b), an NPP model with Granger causality statistic learned. We also considered other widely used parametric/nonparametric TPPs: 1) Hawkes Process with an exponential kernel (HExp) (Lewis & Mohler, 2011); 8) Inhomogeneous Poisson Process (IPP), where the intensity is a sum of $k$ weighted Gaussian kernels: $\hat{\lambda}(t) = \sum_{i=1}^{k} \alpha_i \left(2\pi\sigma_i^2\right)^{-1/2} \exp\left(-(t - c_i)^2 / \sigma_i^2\right)$, where $c_i$ and $\sigma_i$ are fixed center and standard deviation, and $\alpha_i$ is the weight for kernel $i$.

Table 1: Model properties and interpretability comparison of all baselines and TELLER

| Model | RMTPP | NHP | THP | TR-GRU | RPPN | CAUSE | HExp | IPP | **TELLER** |
|---|---|---|---|---|---|---|---|---|---|
| Interpretable | | | | ✓ | | ✓ | ✓ | ✓ | ✓ |
| Flexible | ✓ | ✓ | ✓ | ✓ | ✓ | ✓ | | | ✓ |
| Parsimonious | | | | | | | ✓ | ✓ | ✓ |
| Pairwise interaction | | | | | | ✓ | ✓ | | ✓ |
| Higher-order interaction | | | | | | | | | ✓ |

In Tab. 1, we compared TELLER with these baselines by the model properties: 1) Interpretable: model parameters and prediction are understandable to humans; 2) Flexible: model is expressive to

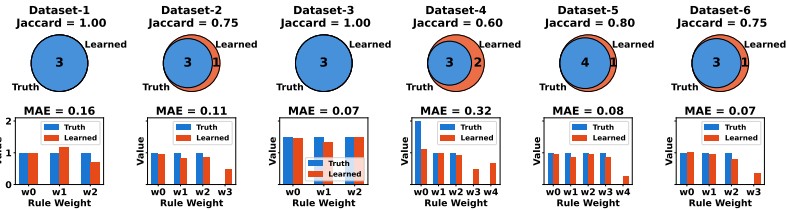

Figure 5: Rule discovery and weight learning results of TELLER-REFS on 6 synthetic datasets (2.4K seqs). capture sophisticated non-linear dependencies; 3) Parsimonious: model parameters are of small or medium size; 4) Pairwise interaction: can identify the mutual influence of the events; 5) Higher-order interaction: can identify the composite effect of a subset of events to another event.

## 4.2 SYNTHETIC DATA

We verify TELLER's rule discovery ability on synthetic datasets with ground truth. The synthetic events are generated from TLPPs with a known set of rules and weights. We prepared 12 synthetic datasets with various rule lengths, weights, and temporal relations. See Appendix G for details. Both the REFS and RAFS subproblem search schemes are tested on all 12 datasets.

Consistent results show an accurate performance of our algorithm in terms of both the rule discovery and parameter learning. We list all the discovered rules based on dataset-1 using 2400 and 600 sequences respectively in Tab. 2, where the correct rules are marked in blue. For dataset-1, the body predicate set is {A, B, C, D}, and the head predicate is E. Given 600 sequences, TELLER successfully discovered 2 out of 3 truth rules along with several noise rules; whereas given 2400 sequences, TELLER accurately uncovered all 3 ground truth rules without any noise rules. The results indicate TELLER is capable of mining rules from noisy event data, and that performance improves with more samples.

Table 2: Learned Rules on Synthetic Dataset1.

| True Rules (# 3) v.s. Learned Rules (REFS) |
|---|
| **2400 Sequences:** |
| E ← A ∧ (A Before E) |
| E ← B ∧ C ∧ (B Before E) ∧ (C Before E) |
| E ← C ∧ D ∧ (C Before D) ∧ (D Equal E) |
| **600 Sequences:** |
| E ← A ∧ (A Before E) |
| E ← A ∧ D ∧ (A Equal D) ∧ (A Before E) |
| E ← A ∧ ¬B ∧ (A Before E) ∧ (¬B Before A) |
| E ← B ∧ C ∧ (B Before E) ∧ (C Before E) |
| E ← B ∧ C ∧ (B Before E) ∧ (C Before B) |
| E ← A ∧ C ∧ (A Before E) ∧ (C Before E) |
| E ← C ∧ A ∧ (A Equal C) ∧ (A Before E) |
| E ← A ∧ ¬C ∧ (A Before ¬C) ∧ (¬C Before E) |

More datasests' results are reported in Fig. 5, where 6 synthetic datasets with 2400 sequences are used for evaluation. Each plot in the top row uses a Venn diagram to show the true rule set and the learned rule set, from which the Jaccard similarity score (area of the intersection divided by the area of their union) is calculated. TELLER discovered almost all the true rules. Each plot in the bottom compares the true rule weights with the learned rule weights, with the Mean Absolute Error (MAE) reported. Almost all truth rule weights are accurately learned and the included noise rule weights are relatively small. In dataset-4, we crafted a long and complex rule with 3 body predicates to challenge TELLER. It is still able to correctly discover this rule, while the weight is slightly underestimated. The existence of long logic rules will require more data and stronger signals (i.e., higher rule weights) to have an accurate recovery. More information about the experiment settings can be found in Appendix G.

We also compared TELLER with the brute-force method, where we enumerated all possible temporal logic rules and built a full model to learn the rule weights. We showed the training process for dataset-1 in Fig. 6, where the evolving likelihood versus the run time is displayed. For TELLER, we also marked the time point when the true rule is discovered (gray dashed vertical lines). The key observations are: 1) For TELLER, the likelihood increases

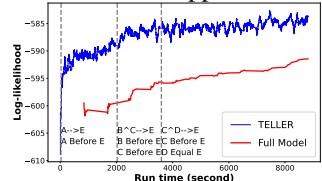

Figure 6: Training process.

very fast and we can see evident jumps whenever a true rule is discovered. As a comparison, the likelihood for the brute-force method increases slowly. 2) All the true rules can be discovered efficiently by TELLER and this verifies that we can set a time limit for early stopping. We can stop TELLER when the objective function seems to start centering around some level. This experiment verifies that our method is computationally efficient compared to the brute-force methods.

## 4.3 REAL DATA: TREATMENT ON MIMIC-III

MIMIC-III contains electronic health records of patients admitted to the intensive care unit (ICU) (Johnson et al., 2016). We focused on patients diagnosed with sepsis (Saria, 2018; Raghu et al., 2017; Peng et al., 2018), as one of the major causes of mortality in ICU. Evidence suggests that the treatment strategy remains uncertain – it is unclear how to use intravenous fluids and vasopressors to support the circulatory system. There also exists clinical controversy about when and how to use these two groups of drugs to reduce the side effect. TELLER is aimed to learn the explanatory temporal logic rules regarding these two groups of drugs as well as their importance weights.

**Predicates and dataset statistics.** We define 62 predicates, including two groups of drugs (i.e., intravenous fluids and vasopressors) and lab measurements. The variables involved are similar to (Saria, 2018). We define two head predicates: 1) LowUrine and 2) Survival. We treat real-time urine as a head predicate since low urine is the direct indicator of bad circulatory systems and is an important signal for septic shock. A complete table of predicates can be found in Appendix H. In our experiment, lab measurement variables are converted to binary values (according to the normal range used in medicine) with the transition times recorded. For drug predicates, they are recorded as 1 when they were applied to the patient. We extracted 4298 patient sequences, and randomly chose 80% of them for training and the remaining for testing. The average time horizon is 392.69 hours and the average #events per sequence is 79.03. Detailed setup can be found in Appendix H.2.

**Evaluation metrics and results.** We use two evaluation metrics: 1) LowUrine is evaluated by predicting the time of transitions from state 0 to state 1 , for which the performance is measured by Mean Absolute Error (MAE). 2) Survival is evaluated by predicting whether the patient is survived at the end of treatment, for which the performance is measured by Accuracy(ACC). Comparison results are in Tab. 3.

Table 3: MIMIC-III: Event prediction results.

| Method | LowUrine (MAE) | Survival (ACC) |
|---|---|---|
| RMTPP | 1.983 | 0.796 |
| NHP | 1.684 | 0.802 |
| THP | 1.545 | 0.824 |
| TR-GRU | 2.666 | 0.811 |
| RPPN | 1.640 | 0.767 |
| CAUSE | 1.712 | 0.740 |
| HExp | 2.578 | 0.882 |
| IPP | 2.472 | 0.784 |
| **TELLER** | **1.266** | **0.930** |

**Discovered temporal logic rules.** Tab. 4 shows the uncovered explanatory temporal logic rules and weights learned by TELLER. We use LowUrine as the head predicate and NormalUrine as its negation. For each discovered rule, the sign of the head predicate will be automatically determined by the algorithm. Human experts confirmed that these discovered logic rules are consistent with the pathogenesis of sepsis, and have captured the most important factors in affecting real-time urine output and survival. Experts found that Rule 1 to Rule 4 capture the major lab measurements that usually emerge together with extremely low urine. Rule 5 seemed hard to interpret in the beginning by experts, since they thought it was slightly counter-intuitive. They later concluded that, in some extreme cases, even after using Crystalloid ( intravenous fluids) as the treatment, the urine can remain low, and thus Rule 5 is still likely to be observed. Rules 6-9 shed light on drug selection. For example, in Rule 7, using Crystalloid and Phenylephrine (vasopressors) together will play an evident role in triggering the urine to a normal level. In Rules 6, 8, and 9, using Crystalloid yields a weight of 4.14, using Colloid (a type of intravenous fluids) yields a weight of 1.97, and using the combinations of the two yields a weight of 1.83, which can be regarded as an incremental contribution.

**Expert-Model interaction.** Experts are involved in TELLER's learning loop: 1) Experts can provide a predicate set, their interested head predicates, an initial rule set, and desired rule templates to the model; 2) Run TELLER to obtain preliminary results for experts to check. Experts can delete unreasonable or noise rules and give the refined rule set to the model before continuing learning. Experts provided advice for predicates definition and the desired rule templates. We ran TELLER from an empty set and generated 12 rules. Experts verified the rules' correctness and deleted the following ones: LowUrine ← HighSpO2; LowUrine ← NormalSodium; LowUrine ← NormalSodium ∧ Water. We were informed that the first rule is contradictory with Pathophysiology common sense, while the other two rules are correct yet irrelevant to the treatment strategy. Then TELLER is re-implemented to refine the result, as in Tab. 4.

Table 4: Learned Rules with LowUrine as the head predicate.

| Weight | Rule |
|---|---|
| 1.07 | **Rule 1:**LowUrine ← LowSysBP ∧ (LowSysBP Before LowUrine) |
| 0.89 | **Rule 2:**LowUrine ← LowSodim ∧ (LowSodim Before LowUrine) |
| 1.16 | **Rule 3:**LowUrine ← HighCreatinine ∧ Equal(HighCreatinine, LowUrine) |
| 1.98 | **Rule 4:**LowUrine ← HighBUN ∧ (HighBUN Before LowUrine) |
| 1.10 | **Rule 5:**LowUrine← HighBUN ∧ Crystalloid ∧ (HighBUN Before LowUrine) ∧ (Crystalloid Before LowUrine) |
| 4.14 | **Rule 6:**NormalUrine ← LowUrine ∧ Crystalloid ∧ (LowUrine Equal NormalUrine) ∧ (Crystalloid Equal NormalUrine) |
| 3.13 | **Rule 7:**NormalUrine ← LowUrine ∧ Phenylephrine ∧ (LowUrine Equal NormalUrine) ∧ (Phenylephrine Equal NormalUrine) |
| 1.83 | **Rule 8:**NormalUrine ← LowUrine ∧ Crystalloid ∧ Colloid ∧ (Colloid Equal NormalUrine) ∧ (Crystalloid Equal NormalUrine) ∧ (LowUrine Equal NormalUrine) |
| 1.97 | **Rule 9:**NormalUrine ← LowUrine ∧ Colloid ∧ (LowUrine Equal NormalUrine) ∧ (Colloid Equal NormalUrine) |

## 5 CONCLUSION

In this paper, we have proposed a new algorithm TELLER to learn interpretable temporal logic rules to explain point processes, with promising results on both synthetic and real-world data. To our best knowledge, this is the first work that automates the discovery of the temporal logic rules based on point process models. Our method adds transparency and interpretability to event models.

ACKNOWLEDGMENTS

This work was in part supported by Shanghai Municipal Science and Technology Major Project (2021SHZDZX0102), Neil Shen's SJTU Medical Research Fund and NSFC (61972250, U19B2035).

Shuang Li's research was in part supported by the Start-up Fund UDF01002191 of The Chinese University of Hong Kong, Shenzhen, Shenzhen Institute of Artificial Intelligence and Robotics for Society, and Shenzhen Science and Technology Program JCYJ20210324120011032

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

## A  RELATED WORK

Our work adds the interpretability to temporal point process models and extends the logic learning method.

**Temporal point process models** have been an elegant tool for event data learning, either for future prediction or (quasi-) causality discovery (Yan et al., 2019; 2016; Liu et al., 2017; Wu et al., 2018b). Traditional parametric models such as Hawkes process (Hawkes, 1971b) are built on simple assumptions such that past events will boost the occurrence of future events. To capture more complicated dynamics, Du et al. (2016a) proposed the first neural point process (NPP) model, where the intensity function is modeled by a Recurrent Neural Network (RNN). Mei & Eisner (2016) improved RMTPP by constructing a continuous-time RNN. Zuo et al. (2020); Zhang et al. (2020a) further leveraged the self-attention mechanism to capture the long-term dependencies of events and meanwhile enhance the computational efficiency. While these neural-based point process models are flexible and excel at event prediction, they are hard to interpret. To add transparency to the black-box models, recently Zhang et al. (2021) introduced Granger causality as a latent graph to explain point processes and the structures are jointly learned via gradient descent. However, Granger causality is still limited to the mutual triggering patterns of events. To incorporate more sophisticated and interpretable event patterns Li et al. (2020) proposed a Temporal Logic Point Process (TLPP), which constructs intensity function by pre-specified temporal logic rules. In fact, such temporal logic rule guided symbolic reasoning process will require highly expressive models such as Transformer to have a good approximate to their solutions (Finkbeiner et al., 2020). However, TLPP can not automatically discover rules and our TELLER overcomes this major drawback.

**Probabilistic logic model** dates back to Markov Logic Networks (MLN) (Richardson & Domingos, 2006; Singla & Domingos, 2008), where a Markov random field is used to model the first-order logic rules and their weights will be learned. Enhanced methods (Papai et al., 2012; Tran & Davis, 2008; Brendel et al., 2011) are generally based on probabilistic graphical models, which allow for partial and noisy observations yet at the cost of intensive computation for inference. TLPP (Li et al., 2020) and our TELLER simplify their setting with two assumptions, i.e. causal direction and fully observed states, and then the temporal logic model can be formulated as a continuous-time point process, which bridges the first-order temporal logic rules to point process.

**Logic learning methods** include SATNet (Wang et al., 2019a), which transforms rule mining into an SDP-relaxed MaxSAT problem, and attention-based methods (Yang & Song, 2019). Some works (Wang et al., 2017; Evans & Grefenstette, 2018; Wei et al., 2019) formulated logic learning as learning an explanatory binary classifier. For example, Evans & Grefenstette (2018) constructed a differentiable model that predicts the conditional probability of the outcome label for a ground atom. Neural-LP (Yang et al., 2017) provided the first fully differentiable rule mining method based on TensorLog (Cohen, 2016), and Wang et al. (2019b) extended Neural-LP to learn rules with numerical values via dynamic programming and cumulative sum operations. In addition, DRUM (Sadeghian et al., 2019) connected learning rule confidence scores with low-rank tensor approximation. Among all these methods, Dash et al. (2018) and Wei et al. (2019) first introduced the Column Generation algorithm in logic learning, which is the closest to our learning framework. However, all these methods are restricted to the static setting and cannot be directly implemented on event sequences.

## B  ALLEN'S THIRTEEN TEMPORAL RELATION

Allen's original paper (Allen, 1990) defined 13 types of *temporal relations between two time intervals*. Specifically, define the time intervals for predicate $x_A$ and predicate $x_B$ as $I_{t_A} = (t_{A_1}, t_{A_2}]$ and $I_{t_B} = (t_{B_1}, t_{B_2}]$ respectively, where $t_{A_1}$ and $t_{B_1}$ are the transition times that the predicate enters the state, and and $t_{A_2}$ and $t_{B_2}$ are the time that the predicate leaves the state. The temporal relation, denoted as $R_{A\ type\ B}(I_{t_A}, I_{t_B})$, is a logic function defined over the time intervals.

See below Table 5 for an illustration. As shown in this table, the temporal relation can be mathematically evaluated by a step function

$$g(s) := \mathbb{1}(s \geq 0)$$

and an indicator function

$$\kappa(s) := \mathbb{1}(s = 0).$$

Considering the inverses of the listed relations plus the symmetric relation "Equal", there are a total of 13 relations.

In practice, to tolerate noise, i.e., the imprecisely recorded time information, it makes sense to introduce softened approximation functions for the step function $g(s)$ and the delta function $\kappa(s)$ in

Table 5: Interval-based temporal relations.

| Temporal Relation | Logic Function $R_{A\ type\ B}(I_{t_A}, I_{t_B})$ | Illustration |
|---|---|---|
| $A$ Before $B$ | $g(t_{B_1} - t_{A_2})$ | |
| $A$ Meets $B$ | $\kappa(t_{A_2} - t_{B_1})$ | |
| $A$ Overlaps $B$ | $g(t_{B_1} - t_{A_1}) \cdot g(t_{B_1} - t_{A_2}) \cdot g(t_{B_2} - t_{A_2})$ | |
| $A$ Starts $B$ | $\kappa(t_{A_1} - t_{B_1}) \cdot g(t_{B_2} - t_{A_2})$ | |
| $A$ Contains $B$ | $g(t_{B_1} - t_{A_1}) \cdot g(t_{A_2} - t_{B_2})$ | |
| $A$ Finished-by $B$ | $g(t_{B_1} - t_{A_1}) \cdot \kappa(t_{A_2} - t_{B_2})$ | |
| $A$ Equals $B$ | $\kappa(t_{A_1} - t_{B_1}) \cdot \kappa(t_{A_2} - t_{B_2})$ | |

replacement of those used in the definitions of temporal relations in Table 5. Step function $g(s)$ can be softened as, e.g., a triangular function or a logistic function, i.e.,

$$g(s) = \min(1, \max(0, \beta s + \tfrac{1}{2})),$$

$$\text{or} \quad g(s) = \frac{1}{1 + \exp(-\beta s)}. \tag{11}$$

Delta function $\kappa(s)$ can be softened as a triangular function or a Laplace density function, i.e.,

$$\kappa(s) = \max(0, \min(\tfrac{s}{\gamma^2} + \tfrac{1}{\gamma}, -\tfrac{s}{\gamma^2} + \tfrac{1}{\gamma})),$$

$$\text{or} \quad \kappa(s) = \frac{\exp(-|s|/\gamma)}{\gamma}. \tag{12}$$

Decaying parameters $\beta$ and $\gamma \geq 1$ can be pre-specified or treated as unknown parameters, which can be learned from data by maximizing the likelihood. In this paper, we pre-specify these decaying parameters and make them frozen in the learning process (i.e., both the master and the subproblem).

## C  PROOF OF THE LIKELIHOOD

Given a realization of all predicates $\{\mathcal{H}(t)\}_{0<t<T}$, one can write out the likelihood function in terms of the intensity function as follows.

For the head predicate $Y$, denote the dual conditional intensity function as $\lambda^*(t)$ and $\mu^*(t)$. Let $p(t_{n+1}|\mathcal{H}_{t_n}, y(t_n) = 0)$ and $p(t_{n+1}|\mathcal{H}_{t_n}, y(t_n) = 1)$ be the *conditional density function* of the next event time $t_{n+1}$ given history and $y(t_n) = 0$, $y(t_n) = 1$, respectively. Let $F(t|\mathcal{H}_{t_n}, y(t_n) = 0)$ and $F(t|\mathcal{H}_{t_n}, y(t_n) = 1)$ be the corresponding cumulative distribution function for any $t > t_n$.

Based on the definition of the conditional transition intensity (or called hazard function), we have

$$\lambda^*(t) = \frac{p(t|\mathcal{H}_{t_n}, y(t_n) = 0)}{1 - F(t|\mathcal{H}_t, y(t_n) = 0)},$$

$$\text{and} \quad \mu^*(t) = \frac{p(t|\mathcal{H}_{t_n}, y(t_n) = 1)}{1 - F(t|\mathcal{H}_{t_n}, y(t_n) = 1)}. \tag{13}$$

From (13), we have

$$\lambda^*(t) = -\frac{d}{dt} \log(1 - F(t|\mathcal{H}_t, y(t_n) = 0)),$$

$$\mu^*(t) = -\frac{d}{dt} \log(1 - F(t|\mathcal{H}_t, y(t_n) = 1)).$$

Integrating both sides, we can get the conditional density and the cumulative distribution function,

$$p(t|\mathcal{H}_{t_n}, y(t_n) = 0) = \lambda^*(t) \exp\left(-\int_{t_n}^t \lambda^*(s)ds\right),$$

$$F(t|\mathcal{H}_{t_n}, y(t_n) = 0) = 1 - \exp\left(-\int_{t_n}^t \lambda^*(s)ds\right),$$

$$p(t|\mathcal{H}_{t_n}, y(t_n) = 1) = \mu^*(t) \exp\left(-\int_{t_n}^t \mu^*(s)ds\right),$$

$$F(t|\mathcal{H}_{t_n}, y(t_n) = 1) = 1 - \exp\left(-\int_{t_n}^t \mu^*(s)ds\right).$$

Let $t_0 = 0$. Given the transition times $(t_1, t_2, \ldots, t_n)$, and suppose $y(t_0) = 0$, $y(t_n) = 1$ and the head predicate is still in state 1 at time $T$, the likelihood function can be factorized into all the conditional densities of each points given all points before it, i.e., the likelihood function is
$\mathcal{L} = p(t_1 | \mathcal{H}_{t_0}, y(t_0) = 0) p(t_2 | \mathcal{H}_{t_1}, y(t_1) = 1) \cdots p(t_n | \mathcal{H}_{t_{n-1}}, y(t_{n-1}) = 0)(1 - F(t | \mathcal{H}_{t_n}, y(t_n) = 1))$.
Plugging in the conditional density function and the cumulative distribution function, the likelihood is expressed as,

$$
\begin{aligned}
\mathcal{L} &= \lambda^*(t_1) \exp\left(-\int_0^{t_1} \lambda^*(s) ds\right) \cdot \mu^*(t_2) \exp\left(-\int_{t_1}^{t_2} \mu^*(s) ds\right) \\
&\quad \cdots \lambda^*(t_n) \exp\left(-\int_{t_{n-1}}^{t_n} \lambda^*(s) ds\right) \cdot \exp\left(-\int_{t_n}^{t} \mu^*(s) ds\right),
\end{aligned}
$$

which completes the proof.

## D  OPTIMALITY CONDITION AND COMPLEMENTARY SLACKNESS

We will provide more descriptions on the optimality condition and the complementary slackness, which provides a sound guarantee to our learning algorithm.

Given the original restricted convex problem,

$$
\text{P}_{\text{Master}}: \quad \boldsymbol{w}^*, b_0^*, b_1^* = \underset{\boldsymbol{w}, b_0, b_1}{\operatorname{argmin}} -\ell(\boldsymbol{w}, b_0, b_1) + \sum_{f \in \bar{\mathcal{F}}} c_f w_f; \quad s.t. \quad w_f \geq 0, \quad f \in \bar{\mathcal{F}} \quad (14)
$$

where parameter $c_f$ depends on the complexity of rule $f$, such as the number of predicates involved in $f$ (i.e., rule length).

The Lagrangian of the original master problem is

$$
L(\boldsymbol{w}, b_0, b_1, \boldsymbol{\nu}) = -\ell(\boldsymbol{w}, b_0, b_1) + \sum_{f \in \bar{\mathcal{F}}} c_f w_f - \sum_{f \in \bar{\mathcal{F}}} \nu_f w_f, \quad (15)
$$

where $\nu_f \geq 0$ is the Lagrange multiplier associated with the non-negativity constraints of $w_f$. As it is a convex problem and strong duality holds under mild conditions. Define $\boldsymbol{w}^*, b_0^*, b_1^*$ as the primal optimal, and $\boldsymbol{\nu}^*$ as the dual optimal, then:

$$
\begin{aligned}
-\ell(\boldsymbol{w}^*, b_0^*, b_1^*) &= \inf_{\boldsymbol{w}, b_0, b_1} L(\boldsymbol{w}, b_0, b_1, \boldsymbol{\nu}^*) \qquad \text{(strong duality)} \\
&= \inf_{\boldsymbol{w}, b_0, b_1} \left(-\ell(\boldsymbol{w}, b_0, b_1) + \sum_{f \in \bar{\mathcal{F}}} c_f w_f - \sum_{f \in \bar{\mathcal{F}}} \nu_f^* w_f\right) \\
&\leq -\ell(\boldsymbol{w}^*, b_0^*, b_1^*) + \sum_{f \in \bar{\mathcal{F}}} \lambda_f w_f^* - \sum_{f \in \bar{\mathcal{F}}} \nu_f^* w_f^* \\
&\leq -\ell(\boldsymbol{w}^*, b_0^*, b_1^*) + \sum_{f \in \bar{\mathcal{F}}} \lambda_f w_f^*.
\end{aligned} \quad (16)
$$

Therefore, $\sum_{f \in \bar{\mathcal{F}}} \nu_f^* w_f^* = 0$, for $f \in \bar{\mathcal{F}}$. This implies the *complementary slackness*, i.e.,

$$
w_f^* = 0 \Rightarrow \nu_f^* \geq 0, \qquad w_f^* > 0 \Rightarrow \nu_f^* = 0 \quad (17)
$$

Given the Karush-Kuhn-Tucker (KKT) conditions, gradient of Lagrangian $L(\boldsymbol{w}^*, b_0^*, b_1^*, \boldsymbol{\nu}^*)$ w.r.t. $\boldsymbol{w}^*, b_0^*, b_1^*$ vanishes, i.e.,

$$
\nu_f^* := -\left.\frac{\partial \ell(\boldsymbol{w}, b_0, b_1)}{\partial w_f}\right|_{\boldsymbol{w}^*, b_0^*, b_1^*} + c_f. \quad (18)
$$

In summary, combining conditions (17) and (18), we obtain the optimalitiy condition,

1. if $w_f^* > 0$, then $\nu_f^* = 0$;

2. if $w_f^* = 0$, then $\nu_f^* \geq 0$,

where the gradient $\nu_f^*$ can be computed via (18).

# E  ALGORITHM BOX

Our TELLER alternates between solving a restricted master problem and a subproblem. We summarize the algorithm in Algorithm 1 and Algorithm 2. RMP indicates the Restricted Master Problem used to update model parameters. SP refers to the Sub-Problem used to construct a new rule. Here we use RAFS as our search scheme.

---

**Algorithm 1:** TELLER (RAFS)

---

**Input:** TimeLimit, MaxRuleLen
**Output:** ruleSet

1  stack ← empty stack;
2  ruleSet ← empty set;
3  $b \leftarrow 0$;
4  $w \leftarrow 0$;
5  $b, w \leftarrow$ RMP($b$, $w$, ruleSet); // Initialize weights and bases.
6  **while** *RunTime ≤ TimeLimit* **do**
7   **if** *stack.isEmpty()* **then**
8    NewRule ← SP($b$, $w$, ruleSet, None); // Search simple rule with length = 1.
9    **if** *NewRule is None* **then**
10    break; // If simple rule does not exist, algorithm ends.
11  **else**
12   RuleToExtend ← stack.top();
13   NewRule ← SP($b$, $w$, ruleSet, RuleToExtend);// Try to extend this rule.
14   **if** *NewRule is None* **then**
15    stack ← empty stack;// If this rule can not be extended, do not revisit it.
16   **if** *len(NewRule)=MaxRuleLen* **then**
17    stack ← empty stack;// If this rule reaches the maximum rule length, stop extending it.
18  **if** *NewRule* **then**
19   ruleSet.add(NewRule);
20   stack.push(NewRule);
21   $b, w \leftarrow$ RMP($b$, $w$, ruleSet);// After adding new rule, update weights and bases
22 **return** *ruleSet*

---

---

**Algorithm 2:** SubProblem (SP)

---

**Input:** $b, w$, ruleSet, RuleToExtend
**Output:** NewRule

1  TempRelSet $\leftarrow \{R_{\text{be}}, R_{\text{eq}}, R_{\text{me}}, \ldots, \text{Null}\}$; // Thirteen types of temporal
   relation and Null (i.e., no temporal relation constraint).
2  bodyPredSet, Y $\leftarrow$ defined by dataset;
3  candRuleSet $\leftarrow$ empty set;
4  **if** RuleToExtend is None **then**
      // Search simple rule with length = 1.
5     **for** $\text{sign}(Y)$ in $\{+, -\}$ **do**
6        **for** $X$ in bodyPredSet and $\text{sign}(X)$ in $\{+, -\}$ **do**
7           **for** $R_{X,Y}$ in TempRelSet **do**
8              candRuleSet.add($(\neg)Y \leftarrow (\neg)X \wedge R_{X,Y}$);

9  **else**
      // Try to extend input rule.
10    **for** $X$ in bodyPredSet.difference(RuleToExtend) and for $\text{sign}(X)$ in $\{+, -\}$ **do**
11       **for** $X'$ in RuleToExtend **do**
12          **for** $R_{X,X'}$ in TempRelSet **do**
13             candRuleSet.add($(\neg)Y \leftarrow$ RuleToExtend $\wedge (\neg)X$ $(\wedge_{X'\text{in RuleToExtend}} R_{X,X'})$);

14 optValue, optRule $\leftarrow$ Evaluate the subproblem objective and choose the smallest one from the
   candRuleSet;
15 **if** optValue $< 0$ **then**
      // This rule is a valid rule, i.e.  it can improve RMP.
16    NewRule $\leftarrow$ optRule;
17 **else**
18    NewRule $\leftarrow$ None;
19 **return** NewRule

---

## F   Search Scheme for Subproblem: Rule-Addition-First Search

The search scheme can also be the Rule-Addition-First Search (RAFS). For the RAFS, the algorithm
starts with an empty rule set and first focuses on discovering all rules with only one body predicate.
Then assign each of them a score that equals the objective function of the subproblem (pricing
problem) applied to the rule. To expand the rules from length $l$ to $l + 1$, we do the following: we
process all generated rules that have $l$ predicates in increasing order of their score (since we aim to
find a negative reduced cost), and for each such rule, we create new rules by appending an additional
predicate together with the associated temporal relations. Whenever we find a rule with a negative
reduced cost, we add it to the current rule list. When our enumeration terminates, we return the best
rules generated by the heuristic before proceeding to the next value of $l$. When none of the rules with
length $l$ can be extended to $l + 1$, we proceed to expand rule from length $l + 1$ to $l + 2$. We may also
pre-specify a maximum rule length and set a time limit to the algorithm.

# G    Synthetic Experiment Results

**Dataset description**: For synthetic experiment, we systematically considered 12 settings, where settings-$\{1, 7, 9, 10, 11, 12\}$ are reported as dataset-$\{1, 2, 3, 4, 5, 6\}$ in main text Section 4.2. Each setting corresponds to different rule weights, rule length and number, type of temporal relation, and intensity of free predicates.

To verify the similarity between the ground truth rules and the generated rules, we further utilize the Jaccard coefficient to measure the degree of consistency between the generated rules and the truth rules. For the $i$-th ground truth rule data, let $\hat{U}_i$ be the set of rules in the $i$-th ground truth rule and $U_i$ be the $i$-th rules generated by our method. The Jaccard is defined as $\frac{|U_i \cap \hat{U}_i|}{|U_i \cup \hat{U}_i|}$.

The Jaccard similarities (range from 0 to 1) are reported in Table 6 and Fig. 7. We explore how the sample size (600, 1200, 2400) and the search method (REFS and RAFS) will impact the performance. We plot the Jaccard of our method in different sample sizes and search methods on 12 settings (see Table 7 for descriptions). The results are summarized in Fig. 7. We find that the performance gradually improves with the increase of sample size, which verifies that sufficient data can benefit the learning performance.

As mentioned in Section 3.3, REFS will always extend the longest existing rule, and RAFS is always extending the shortest existing rule. As shown in Fig. 7, REFS achieves similar performance with RAFS on most settings.

Table 6: Synthetic Data: Jaccard similarity with the Ground Truth Rules.

| Setting | REFS-600 | REFS-1200 | REFS-2400 | RAFS-600 | RAFS-1200 | RAFS-2400 |
|---|---|---|---|---|---|---|
| Setting-1 | 0.222 | 0.972 | 1.000 | 0.200 | 1.000 | 0.500 |
| Setting-2 | 0.059 | 0.000 | 0.250 | 0.200 | 0.200 | 0.222 |
| Setting-3 | 0.182 | 0.286 | 0.750 | 0.286 | 0.375 | 0.750 |
| Setting-4 | 0.111 | 0.429 | 0.600 | 0.117 | 0.428 | 0.600 |
| Setting-5 | 0.200 | 0.571 | 1.000 | 0.125 | 0.500 | 1.000 |
| Setting-6 | 0.250 | 0.667 | 0.750 | 0.222 | 0.333 | 0.600 |
| Setting-7 | 0.250 | 0.750 | 0.750 | 0.090 | 0.750 | 0.600 |
| Setting-8 | 0.167 | 0.000 | 0.286 | 0.071 | 0.00 | 0.200 |
| Setting-9 | 0.750 | 0.600 | 1.000 | 0.600 | 0.750 | 1.000 |
| Setting-10 | 0.059 | 0.231 | 0.600 | 0.111 | 0.154 | 0.750 |
| Setting-11 | 0.444 | 0.571 | 0.800 | 0.364 | 0.444 | 0.800 |
| Setting-12 | 0.375 | 0.429 | 0.750 | 0.091 | 0.429 | 1.000 |

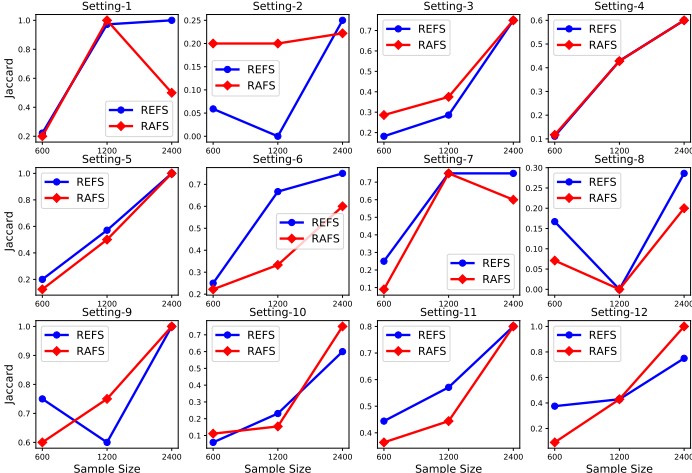

Figure 7: Jaccard similarity on Synthetic Data: with REFS Results in Blue and RAFS Results in Red.

Table 7: Synthetic Data Setting and Observations.

| Dataset | Observation |
|---|---|
| **Setting-1: Same body predicate intensity**
$w = [1, 1, 1]$
$\lambda = [1, 1, 1, 1]$ | Hard scenario since all body predicates have the same occurrence rate. TELLER can still recover the rules (Table 2) and their weights. |
| **Setting-2: Low important weights**
$w = [0.5, 0.5, 0.5]$
$\lambda = [1, 1, 1, 1]$ | Lowering the important weights enhances the challenges in learning. |
| **Setting-3: High important weights**
$w = [1.5, 1.5, 1.5]$
$\lambda = [1, 1, 1, 1]$ | For relatively high rule weights, TELLER can accurately recover the rules and their weights. |
| **Setting-4: Add rule length**
$w = [1, 1, 2]$
$\lambda = [1, 1, 1, 1]$ | Adding rule length enhances the challenges in learning. |
| **Setting-5: Add one rule**
$w = [1, 1, 1, 1]$
$\lambda = [1, 1, 1, 1]$ | Adding one rule does not harm the rule recovery and learning performance. |
| **Setting-6: Add one dummy predicate**
$w = [1, 1, 1, 1]$
$\lambda = [1, 1, 1, 1, 0.2]$ | Adding dummy predicates enhances the challenges in learning. |
| **Setting-7: Different body predicate intensities**
$w = [1, 1, 1]$
$\lambda = [0.6, 0.8, 1.2, 1.4]$ | Body predicates have different occurrence rates. The MAE results outperform **Setting-1** |
| **Setting-8: Different intensities and low weights**
$w = [0.5, 0.5, 0.5]$
$\lambda = [0.6, 0.8, 1.2, 1.4]$ | Lowering the important weights enhances the challenges in learning, but different intensities lower the challenges. |
| **Setting-9: Different intensities and high weights**
$w = [1.5, 1.5, 1.5]$
$\lambda = [0.6, 0.8, 1.2, 1.4]$ | Both high weights and different intensities lower the challenges. |
| **Setting-10: Different intensities and add rule length**
$w = [1, 1, 2]$
$\lambda = [0.6, 0.8, 1.2, 1.4]$ | Both adding rule length and different intensities enhance the challenges. |
| **Setting-11: Different intensities and add one rule**
$w = [1, 1, 1, 1]$
$\lambda = [0.6, 0.8, 1.2, 1.4]$ | Both adding one rule and different intensities lower the challenges. |
| **Setting-12: Different intensities and add one dummy predicate**
$w = [1, 1, 1]$
$\lambda = [0.6, 0.8, 1.2, 1.4, 0.2]$ | Adding dummy predicate enhances the challenges in learning, but different intensities lower the challenges. |

# H   REAL EXPERIMENT: MIMIC

## H.1   SUPPLEMENTAL INFORMATION

MIMIC-III is a dataset released under PhysioNet Credentialed Health Data License 1.5.0[2]. It was approved by the Institutional Review Boards of Beth Israel Deaconess Medical Center (Boston, MA) and the Massachusetts Institute of Technology (Cambridge, MA). The requirement for individual patient consent was waived because all the patient health information was deidentified. We manually checked that this data do not contain personally identifiable information or offensive content.

## H.2   HYPERPARAMETERS AND EXPERIMENT ENVIRONMENT

For the MIMIC dataset, we limit the maximum rule length to be 3, and the maximum #rules to be 20. The learning rate in solving the restricted master problem is $\times 10^{-4}$. The master problem is optimized by the SGD type of algorithm and we choose to use the projected gradient descent to take care of the weight constraints. The batch size is 64. Each time we solve the subproblem, we randomly selected 50% of the training data (i.e., patient sequences) to evaluate the subproblem objective. To exclude noisy and irrelevant rules, we clip the learned weights and discard these rules with weights smaller than is $10^{-2}$ in solving the restricted master problem. To further reduce the number of candidate rules, we set a threshold to the subproblem gain as $5 \times 10^{-3}$, i.e., we only include the candidate rules with negative cost smaller than $-5 \times 10^{-3}$. Our model is trained and evaluated using 16 processes in parallel, on a server with a Xeon W-3175X CPU.

For the neural-based baselines, we set: 1) TR-GRU with 4 hidden states and 4009 trainable weights, 2) RPPN with 64 hidden states, 64 embedding dimensions, and 33277 trainable weights, 3) CAUSE with 16 hidden states and 19312 trainable weights, 4) NHP with 8 hidden states, 814 trainable weights, 5)THP with 8 hidden states, 3608 trainable weights, 6) RMTPP with 32 hidden states, 5440 trainable weights, and 7) IPP with 3 Gaussian kernels, $\mathbf{c} = [1, 1, 1], \sigma = [1, 0.8, 0.5]$.

## H.3   PREDICATE DEFINITION IN MIMIC

## H.4   DISCOVERED TEMPORAL LOGIC RULES

The discovered rules to explain the real-time urine have been reported in Section 4 Table 4. We list the discovered rules relate to the survival condition here in Table 9.

---

[2]https://physionet.org/content/mimiciii/view-license/1.4/

Table 8: Defined Predicates in Our MIMIC-III Experiment.

| Lab Measurements | Low/Normal/High-SysBP |
|---|---|
| | Low/Normal/High-SpO2SaO2 |
| | Low/Normal/High-CVP |
| | Low/Normal/High-SVR |
| | Low/Normal/High-Potassium |
| | Low/Normal/High-Sodium |
| | Low/Normal/High-Chloride |
| | Low/Normal/High-BUN |
| | Low/Normal/High-Creatinine |
| | Low/Normal/High-CRP |
| | Low/Normal/High-RBCcount |
| | Low/Normal/High-WBCcount |
| | Low/Normal/High-ArterialpH |
| | Low/Normal/High-ArterialBE |
| | Low/Normal/High-ArterialLactete |
| | Low/Normal/High-HCO3 |
| | Low/Normal/High-SvO2ScvO2 |
| Output | LowUrine |
| Input | Colloid, Crystalloid, Water |
| Drugs | Norepinephrine, Epinephrine, Dobutamine, Dopamine, Phenylephrine |
| Suvival Condition | Survival |
| Temporal Relation Type | Before, Equal |

Table 9: Learned Rules with `Survival` head predicate for Sepsis Patients in MIMIC-III

| Weight | Rule |
|---|---|
| 0.95 | **Rule 1:** NotSurvival ← NormalSVR ∧ Epinephrine ∧ (Epinephrine Before NotSurvival) ∧ (NormalSVR Before NotSurvival) |
| 0.91 | **Rule 2:** NotSurvival ← HighArterialBe ∧ (HighArterialBe Before NotSurvival) |
| 0.82 | **Rule 3:** NotSurvival ← ∧ HighBUN ∧ Phenylephrine ∧ (Phenylephrine Before NotSurvival) ∧ (HighBUN Before NotSurvival) |
| 0.51 | **Rule 4:** NotSurvival ← HighSodium ∧ (HighSodium Before NotSurvival) |
| 0.53 | **Rule 5:** NotSurvival ← HighSodium ∧ Norepinephrine ∧ (HighSodium Before NotSurvival) ∧ (Norepinephrine Before NotSurvival) |
| 0.89 | **Rule 6:** Survival ← NormalAterialPH ∧ (NormalAterialPH Before Survival) |
| 0.55 | **Rule 7:** NotSurvival ← HighPotassium ∧ (HighPotassium Before NotSurvival) |
| 1.19 | **Rule 8:** NotSurvival ← ∧ HighPotassium ∧ Colloid ∧ (HighPotassium Before NotSurvival) ∧ (Colloid Before NotSurvival) |
| 1.00 | **Rule 9:** NotSurvival ← HighlAterialPH ∧ UseNorepinephrine ∧ (HighlAterialPH Before NotSurvival) ∧ (HighlAterialPH Before NotSurvival) |
| 0.61 | **Rule 10:** NotSurvival ← HighHCO3 ∧ (HighHCO3 Before Survival) |

# I    REAL EXPERIMENT: CRIME

Crime Incident Reports are provided by Boston Police Department to document the type of each incident as well as when and where it occurred (of Innovation & Technology, 2015).

## I.1    SUPPLEMENTAL INFORMATION

This dataset is released by the Boston Department of Innovation and Technology under Open Data Commons Public Domain Dedication and License (PDDL). The publisher does not discuss how the data was collected and whether consent was obtained. We manually checked that this data does not contain personally identifiable information or offensive content.

**Predicate Definition and Dataset statistics.** We are interested in the top four most frequent crime types: vandalism, theft from motor vehicles, assault, and shoplifting. Another ten predicates are defined to describe the occurrence time properties, such as whether it is in the morning or the afternoon, on weekdays or weekends, in spring or winter, etc. The set of defined predicates is displayed in Table 10. We consider all the crime reports from June 2015 to May 2021 and split the data into 1879 sequences according to days. We randomly choose 80% of these sequences as training data and the remaining as testing data. On average, each sequence contains 46.03 events.

Table 10:  Defined Predicates for Crime.

| Period of Crime | Spring, Summer, Autumn, Winter, Weekday, Weekend, Morning, Afternoon, Evening, Night |
|---|---|
| Crime Types | Vandalism, TheftFromMV, Assault, Shoplifting |
| Temporal Relation Type | Before, Equal |

## I.2    HYPERPARAMETERS AND EXPERIMENT ENVIRONMENT.

Our model is trained and evaluated using 16 processes in parallel, on a server with a Xeon W-3175X CPU. For this Crime dataset, we limit the maximum rule length to be 2, and the maximum #rules to be 20. The learning rate used in updating model parameters in the restricted master problem is $10^{-4}$. The master problem is optimized by SGD with a batch size of 64. The subproblem objective function is evaluated on the entire training data. To exclude noisy and irrelevant rules, we clip the learned weights and discard these rules with weights smaller than is $10^{-2}$ in solving the restricted master problem. To further reduce the number of candidate rules, we set a threshold to the subproblem gain as $\times 10^{-2}$, i.e., we only include the candidate rules with negative cost smaller than $- \times 10^{-2}$.

For the non-parametric baselines, we set: 1) TR-GRU with 4 hidden states and 1129 trainable weights, 2) RPPN with 64 hidden states, 64 embedding dimensions, and 27037 trainable weights, 3) CAUSE with 16 hidden states and 5872 trainable weights, 4) NHP with 8 hidden states, 382 trainable weights, 5)THP with 8 hidden states, 2408 trainable weights, 6)RMTPP with 32 hidden states, 2240 trainable weights, and 7)IPP with 3 Gaussian kernels, $\mathbf{c} = [1, 1, 1], \sigma = [1, 0.8, 0.5]$.

## I.3    PREDICTION ACCURACY

Table 11: Crime: MAE of Event Time Prediction.

| Method | Vandalism | Larceny TheftFromMV | Assault | Larceny Shoplifting |
|---|---|---|---|---|
| RPPN | 0.881 | 1.137 | 1.185 | 0.777 |
| HExp | 0.761 | 0.949 | 1.912 | **0.704** |
| TR-GRU | 0.759 | 1.351 | 1.400 | 1.092 |
| CAUSE | 0.962 | 1.127 | 1.206 | 0.892 |
| NHP | **0.613** | 1.300 | 1.887 | 1.269 |
| THP | 0.973 | 1.043 | **0.957** | 0.939 |
| RMTPP | 0.874 | 1.021 | 1.059 | 0.763 |
| IPP | 0.908 | 1.274 | 1.508 | 1.179 |
| TELLER | 0.770 | **0.826** | 1.465 | 0.710 |

Table 12: Temporal Logic Rules Discovered for Four Types of Crime Events.

| Weight | Rule |
|--------|------|
| 3.84 | **Rule 1:** Vandalism ← Shoplifting ∧(Shoplifting Before Vandalism) |
| 1.83 | **Rule 2:** Vandalism ← TheftFromMV ∧(TheftFromMV Before Vandalism) |
| 0.43 | **Rule 3:** Vandalism ← TheftFromMV ∧ Shoplifting ∧(Shoplifting Before Vandalism) ∧(TheftFromMV Before Vandalism) |
| 2.46 | **Rule 4:** TheftFromMV ← Shoplifting∧ (Shoplifting Before TheftFromMV) |
| 1.42 | **Rule 5:** TheftFromMV ← TheftFromMV ∧ Summer ∧(TheftFromMV Equal Summer) ∧(TheftFromMV Before TheftFromMV) |
| 0.40 | **Rule 6:** Assault ← Assault ∧ Weekend ∧(Weekend Equal Assault) ∧(Assault Equal Assault) |
| 3.62 | **Rule 7:** Assault ← Shoplifting ∧ Assault ∧(Shoplifting Before Assault) ∧(Assault Before Assault) |
| 1.67 | **Rule 8:** Shoplifting ← TheftFromMV ∧(TheftFromMV Before Shoplifting) |
| 1.14 | **Rule 9:** Shoplifting ← Shoplifting ∧ Vandalism ∧(Vandalism Before Shoplifting) ∧(Shoplifting Before Shoplifting) |
| 1.30 | **Rule 10:** Shoplifting ← Shoplifting ∧ Weekday ∧(Shoplifting Before Shoplifting) ∧(Weekday Equal Shoplifting) |

The mean absolute error (MAE) of the predicted event times are displayed in Table 11. Our model outperforms other models in predicting TheftFromMV. It has a comparable performance with the Hawkes baseline on the remaining three tasks. We can think of Hawkes process as a special case of our model and this is especially true if we restrict our model to learn short rules.

## I.4  DISCOVERED RULES

We displayed the discovered important rules in Table 12, from which one can observe the following crime patterns. Shoplifting and TheftFromMV may trigger Vandalism. One explanatory reason is that larceny may involve break-in and destruction of security devices (Rule 1, 2, and 3). Shoplifting triggers TheftFromMV. These two events are special types of larceny, and thus they may exhibit similar crime patterns (Rule 4). TheftFromMV exhibits self-exciting (i.e., clustering) patterns in summer. This is may due to that winter is extremely cold in Boston, and summer, however, is a nice period of the year for outdoor activities and thus motor vehicles may pour into the area, which potentially increases the likelihood of theft (Rule 5). Assault shows a self-exciting pattern on weekends. People tend to have more social activities on weekends, which increases the possibility of domestic violence and affray (Rule 6). Shoplifting triggers Assault. This might be explained by the fact that if shoplifting is spotted at the scene, the thief may have a physical conflict with the security guard (Rule 7). TheftFromMV and Vandalism will work together to trigger Shoplifting, which is consistent with our previous observations 1 and 2 (Rule 8 and 9). Shoplifting is self-exiting on weekdays. Stores are busier on weekends than on weekdays, and thus weekdays might be a better choice for shoplifters (Rule 10).

