# OpenReview forum: "Explaining Point Processes by Learning Interpretable Temporal Logic Rules"
_ICLR.cc/2022/Conference — ICLR 2022 Poster_

### Official Review · Reviewer_AiEu · 2021-11-02

**Correctness:** 3
**Technical Novelty And Significance:** 3
**Empirical Novelty And Significance:** 3
**Recommendation:** 6
**Confidence:** 4

**Details Of Ethics Concerns:**

None.

**Main Review:**


1.	Structure and presentation

The paper is not self-contained on the theoretical basis. It especially lacks necessary explanations on the relationship between interpretable temporal logic rules and temporal point processes. From the problem to the output (such as the derived rules in Table 2), the paper does not seem to show any strong dependencies on the temporal point process (TPP) model. Unfortunately, it neither fully demonstrates and constructs the complete problem background nor provides sufficient technical details on the critical process (rule generation, searching/expanding algorithm, expert knowledge templates). Instead, it took a lot of space to define a two-stage solution framework. For me, this is not the most valuable contribution.

Figure 1 tries to convey too many technical details that are difficult for readers to understand, especially when reading Section 1, until it was cited in the discussion of “Search algorithms” in Section 3.1 to show the necessary details of designing the REFS algorithm. Therefore, figure 1 should focus on TELLER’s core idea rather than technical details in the paper’s introduction part.

The situation in Figure 2 is similar, where the author uses the concept of TLPP before its theoretical preliminaries are introduced in the main text. In the discussions of “Logic-informed intensity function”, the author may consider guiding readers to understand the concept and calculation method of intensity function using case studies designed in Figure 2, rather than directly referring to it without any explanation.

The position of Figure 3 is also problematic. It appears in Section 3.1 and is not quoted by any text until the end of Section 3.4.

2.	Novelty and contribution

For the concepts introduced in Section 2, the authors should clarify which are cited and new tools designed by TELLER. More importantly, it is necessary to reflect the differences between the related work built on TPP model and this study.

To my understanding, it is not an outstanding innovation to define rule discovery as a two-stage optimization problem. The most valuable contribution is in designing a sound rule generation or expanding algorithm in the sub-optimization problem (near-optimal).

Although the paper claimed: “Our problem is even more challenging compared to traditional inductive logic programming cases due to the sequential properties of the data and the presence of temporal information.” However, we do not see any theoretical analysis to show how TELLER has reduced the calculation?

Since the event sequences and values are discrete, is this problem challenging to solve in the case of a limited number of basic events (the number of variables considered in the case analysis is small in reality)? That is, how big is the size of the searching space of the $P_Master problem (Equation 4) in real practices?

In response to this issue, the paper skipped several key details when designing the search algorithms in Section 3.3: (1) Why the complexity is O(d)? In such a not-complex solution space, why should we resort to a complicated searching scheme instead of enumerating? (2) When extending the original rule, TELLER considers “the signs and possible temporal relations with the head predicate”. However, the paper did not show specific principles and rules to do this. Many detailed questions need to be explained: Is the extending process following any temporal sequential way? Can it support the negation pattern? Can it discover the loop structure? And logical branch structure? These factors will significantly affect the ability of the extending algorithm to find effective rules

In Section 3.4, the author mentioned that TELLER's scalability is an advantage over brute-force solving algorithms. Although there can be many event sequences in a complex example, what kind of problem will have so many predicates? For example, we see the cases in the experiment part are:
1. Synthetic data: predicate set {A,B,C,D}, head {E}.
2. MIMIC-III:  62 predicates as stated, head {LowUrine, Survival}.

Finally, we can also see from Crime experiments that users still need to define promising predicates to guide the algorithm to discover rules. It seems the distance between reasonable predicates and truth rule is not far away, or we can say that in the case of limited predicates and temporal relations, the best results can be obtained through brute-force enumeration and evaluation. Therefore, the authors should clarify whether there are problem scenarios that need to deal with massive predicates to prove the necessity of designing TELLER.

3.	Technical details

The authors can consider discussing why we need to introduce rule set weights in logic-informed intensity function.

The paper mentioned that “the initial rule set can be an empty set or any pre-defined small set”. Will an accurate pre-defined set contribute to the accuracy of the algorithm? Better to provide some experimental results and analysis.

In the analysis of TELLER’s scalability, the author mentioned: "In practice, setting the time limit is proven to work well and the most important rules are likely to be discovered." This is a key observation in evaluating TELLER. Therefore, it is recommended to design different time window parameters to verify the effect of the algorithm.

4.	The subjectivity of the experimental results

In Table 1, the author gives five properties but does not explain how to decide whether a method is of or not of these properties. Table 1 shows several arbitrary comparison results without seeing any justified process. It is recommended that the authors provide a detailed definition for several fundamental properties such as Parsimonious and Higher-order interaction and justify the results through a case study or statistical results.

5.	Reproducibility

The author did not attach their codes, which affected the evaluation of the paper’s reproducibility.

6.	Miscellaneous and language problems

Through careful check, it is found that the font size of the paper is not consistent. Starting from the "Logic-informed intensity function" section on page 4, the font suddenly becomes smaller. I am not sure whether this meets the ICLR format requirements because the content from Page 4-8 may exceed the limit.

The first paragraph in Section 3.4 is a redundant summary that deserves to be simplified.

Section 1, Paragraph (Para) 2, “These domain knowledge can be summarized as” -> This domain knowledge

Section 1, Para 3, “without the need to discretize the time horizon into bins.” Not clear, what “bins” refers to ?

Page 2, Para 5, “where the number of variables are too large to be considered explicitly.” -> is too large to

Page 3, Para 6, “the body part of the formula indicate the evidence” -> indicates

Page 4, Para 4, “-1 indicates negative effect,” -> a negative effect

Page 4, the last line, “requires enumerating exponentially large set of combinations” -> an exponentially large set

Page 5, Para 3, “the set of variables (the $w_f$s) we are optimizing on”. It is inappropriate to use plural “s” here.

Page 6, Para 6, “This provides performance guarantee for speedup algorithms.” -> a performance guarantee

Page 6, Para 6, “we do not need to consider rules with length exceeding $H$.” -> with a length exceeding $H$

Page 6, Para 6, “we mean if the conjunctions of the input predicates is important” -> are important

Page 8, the last paragraph, “Among all these predicates we are interested in reasoning about two predicates.” Ambiguous, should be separated by commas

Page 9, Para 1, “is a important signal for septic shock” -> an important

Page 9, the last paragraph, “We propose a promising TELLER algorithm to learn interpretale temporal logic rules” -> interpretable


**Summary Of The Paper:**

This paper introduces TELLER, a framework with novel algorithms based on the temporal point process model to discover interpretable temporal logic rules. To this end, the authors designed a “rule generation - evaluation” two-stage way to solve the problem efficiently. The proposed method is evaluated using one synthetic and two real data sets. Experts have also verified that the learned temporal logic rules are interpretable and reflect many critical principles in real practices.

**Summary Of The Review:**

In summary, although the paper presents a solid theoretical discussion and a comprehensive verification, it still has defects in the overall structure, the clarity of innovation, the challenges of the problem scene, and some method details. As a result, I do not recommend accepting the paper even though it shows many merits.

Revision Notes (2021-11-26):

Most of the problems have been appropriately resolved or explained in the revision. The quality of the paper has been significantly improved and exceeded the acceptance threshold. Therefore, I have raised my score to 6.

---

> ### Author Response · Authors · 2021-11-22
> **In response to reviewer AiEu**
>
> We especially thank you for your careful reading. We are grateful that you proposed so many constructive suggestions.
> The proposed comments are informative and valuable. All your suggestions will help us further improve the quality of this paper.
>
> We will address your concerns from the following aspects.
>
>  **As for the structure and presentation.**
>
> **Q1** Relationship between interpretable temporal logic rules and temporal point processes.
>
> We revised the draft to further clarify the relationship. Please refer to the introduction for the changes.
>
> To summarize, temporal point processes (TPP) is a well-developed continuous-time model for event sequences, but most TPP models lack interpretability. Our discovered temporal logic rules will help "explain" the occurrence of events and add the sequential model's interpretability. On the other hand, treating temporal logic rules as hard constraints will be too strict for recurrent noisy events. Our point process model uses a weighted combination of temporal logic rules to construct the occurrence rate function of the events, which "softened" these logic rules. The probabilistic model allows us to deal with the uncertainty of these irregularly happened events.
>
>
> **Q 2**  Too much information in figures.
>
> As suggested by you, we introduced a simple figure 1. Put the original figure 1 to section 3. We elaborated on the case study discussed in Figure 2. We repositioned the original Figure 3 as well.
>
> **Q 3** Typos
>
> We fixed all your proposed typos and carefully proofread the revised draft.
>
> **Q4** Potential page limit issue
>
> Thank you so much for noticing this! We didn’t realize it until you pointed it out. Now we revised the draft by pruning some redundant descriptions as you suggested and now the page limit should be satisfied.
>
> **As for the novelty and contribution**
>
> **Q 5** Clarify the novelty
>
> Our contribution is being the first work to learn the interval-based temporal logic rules based on temporal point processes. The designed two-state solution framework is an effective and efficient algorithm to help us achieve our goals.
>
> As also suggested by the reviewer “the most valuable contribution is in designing a sound rule generation or expanding algorithm in the sub-optimization problem (near-optimal).” In solving the sub-problem, we leverage the sparsity and heredity principles of the rule sets, as well as the expert’s preference (if accessible) to squeeze the search space and speed up the rule search algorithm.
>
> We also added more experiments as suggested by the reviewer to justify the contributions of our method. More details will be provided below.
>
>
> **Q 6** Clarify which tools are cited
>
> Modeling the intensity function of point processes by temporal logic rules is existing work:
>
> [1] Li, Shuang, Lu Wang, Ruizhi Zhang, Xiaofu Chang, Xuqin Liu, Yao Xie, Yuan Qi, and Le Song. "Temporal Logic Point Processes." ICML, 2020.
>
> However, it is not clear how to uncover these temporal logic rules from event data.
>
> The two-stage column generation algorithm is existing work. For example,
>
> [2] Dash, Sanjeeb, Oktay Günlük, and Dennis Wei. "Boolean decision rules via column generation." NeurIPS 2018.
>
> However, it is not clear how to apply the two-stage column generation algorithm to event sequences to uncover temporal logic rules.

---

> > ### Author Response · Authors · 2021-11-22
> > **More experiments as suggested by the reviewer AiEu**
> >
> > We also added more experiments as suggested by the reviewer to justify the contributions of our method. More details will be provided below.
> >
> > **Q7** How are the computational results compared to the brute-force methods?
> >
> > As suggested by the reviewer, we compared TELLER with the brute-force search method. For the brute-force method, we enumerate all possible temporal logic rules and pack them together to build the full models to learn the rule weights.
> >
> > We did the comparison using our synthetic data example: Our predicate set {A,B,C,D}, head {E}.
> >
> > In this case, it is still durable to enumerate all rules to build a full model. First, consider rules with only one body predicate. There are $4\choose 1$ options. By considering that each predicate can take positive or negative signs, there are $2\times 2$ combinations of the signs. So there are $4\choose 1$$ \times 2\times 2$ options. By enumerating possible temporal relations among the selected predicates, and suppose we only consider Before, Equal (i.e., happens almost the same time), two types of relations, there are $4\choose 1 $ $\times 2\times 2 \times 2$ options.
> >
> > Next, we continue to enumerate rules with two body predicates. Similarly, there are $4\choose 2$ options. By considering each predicate can take positive or negative signs, there are $2\times 2\times 2$ combinations of the signs. So there are $4\choose 2 $$\times 2\times 2 \times $ options. We need to further consider all the pairwise temporal relations of the selected predicates. There are $3\times 2$ possible temporal relations. So there will be $4\choose 2$$ \times 2\times 2 \times \times 3\times 2 $ options. We can proceed with the enumerations until the maximal rule length reaches 3.
> >
> > For this simple case, there are around  592 possible rules (counted by our codes). In a real-world dataset like the MIMIC with 59 body predicates, and again rule length limit is 3, the number of rules is more than 200,000. This is horrible for a naive brute-force model.
> >
> > Experiment setup:
> >
> > For both TELLER and the brute-force method, we visualized the evolving likelihood function versus the run time in the training process. For TELLER, we also marked the time point that the true rule is discovered. The complete experiment result is displayed in the draft. See Fig 5 for details.  Here, we can only report several values from the curve.
> >
> > TELLER's rule                    | Generate time (second) | TELLER Log Likelihood | Full Model Log Likelihood
> >
> > E<—A^(A Before E)                 | 27.96        | -596.20              | No Result
> >
> > E<—B^C^(B Before E）^（C Before E) | 2027.20      | -590.83              |-599.02
> >
> > E<—C^D^(C Before E）^（D Equal E)  | 3589.07      | -587.14              |-597.27
> >
> > There are several key observations:
> >
> > 1. For TELLER, the likelihood function increases very fast at the very beginning of the training process. We can see evident jumps in likelihood whenever a true rule is discovered by TELLER. As a comparison, the likelihood for the brute-force method increases slowly.
> >
> > 2. All the true rules can be discovered efficiently by TELLER and then the likelihood seems to grow slowly. This verifies that a time limit window can be used for TELLER for early stopping. In practice, we can stop TELLER when the objective function seems to start reducing the speed of the growth.
> >
> > This experiment verifies that our method is computationally efficient compared to the brute force methods.

---

> > > ### Author Response · Authors · 2021-11-22
> > > **In response to the remaining questions asked by reviewer AiEu**
> > >
> > > **Q 8** As for the initialization of the subset $\mathcal{F}_0$
> > >
> > > We've answered the same question to reviewer BWwC. For your convenience, we copied the content below.
> > >
> > > We empirically checked this when we were doing the synthetic experiments. For example, we tried randomly generating a short rule with only one body condition to initialize $\mathcal{F}_0$. The choice of $\mathcal{F}_0$ will not influence the final solution that much but will impact the run time.
> > >
> > > A rule of thumb we discovered is to start with an “empty” $\mathcal{F}_0$, which refers to there being only a base term in the intensity function, i.e., intensity = exp(b). The initial estimate of b will be learned by MLE (i.e., restricted master problem). Then we can start adding a new candidate rule by the subproblem.
> > >
> > > In real applications, we can imagine TELLER can leverage domain knowledge to initialize $\mathcal{F}_0$. There is no need to learn known rules from scratch, although we didn't try this initialization in our experiments.
> > >
> > > We have also added the above discussions to section 3.4.
> > >
> > > **Q9** Definitions of parsimonious models and higher-order interaction.
> > >
> > > By “Parsimonious”, we mean the model has a limited number of parameters. Usually, we define a model as parsimonious if its number of parameters is comparable to the dimension of its input.
> > >
> > > By “Higher-order interaction”, we mean the model has an interpretation of the joint effect of 2 or more input variables on the target variables.
> > >
> > > **Q10** Why do we learn the rule weights?
> > >
> > > To convert the temporal logic rules from "hard" constraints to "soft" constraints. The weights can be interpreted as the importance (or confidence level) of each rule.
> > >
> > > **Q11** Show some applications where we have many predicates.
> > >
> > > As the healthcare example considered in this paper, the number of predicates is around 60, defining different symptoms, measurements, treatments to the patients. This is a small set of predicates after doctors' labeling. The total number of predicates in this real problem, without doctors' manual feature selection, will be much bigger than 60.
> > >
> > >
> > > **Q12**  Why the complexity is O(d)? In such a not-complex solution space, why should we resort to a complicated searching scheme instead of enumerating?
> > >
> > > The complexity for each sub-problem is O(d) (note this is not for the entire problem). This is because to solve the subproblem we only choose one predicate from the predicate set to add to the existing rule as an expansion.
> > >
> > > **Q13** Codes are not attached.
> > >
> > > We will release our codes once the paper is published.

---

> > > > ### Author Response · Authors · 2021-11-24
> > > > **In response to reviewer AiEu**
> > > >
> > > > Thank you for all your constructive comments! We wonder whether our response has addressed your major concerns to allow the paper to cross your acceptance threshold. If you have any additional specific concerns, please let us know, and we would be happy to answer them or address them in the final version.

---

> > > > > ### Comment · Reviewer_AiEu · 2021-11-26
> > > > > **Response to the authors' rebuttal**
> > > > >
> > > > > I appreciate the author's meticulous explanation and revision. Most of the problems have been appropriately resolved or explained. I believe that the quality of the paper has been significantly improved and exceeded the acceptance threshold. I will raise my score to 6, primarily because I think it's a critical problem that needs more researchers working on it. I'm inclined to overlook some shortcomings in light of the anticipated effort to put TELLER to larger-scale real-world problems and present inspiring effects.

---

### Official Review · Reviewer_GzHz · 2021-11-04

**Correctness:** 3
**Technical Novelty And Significance:** 3
**Empirical Novelty And Significance:** 3
**Recommendation:** 6
**Confidence:** 2

**Main Review:**

Strength:
* The problem of learning temporal logical rules is very important and interesting.
* The key idea is novel and interesting.
* The experiments are extensive, which are conducted on both synthetic data and real data.
* There are many case studies and learned rules, which is interesting.

Weakness:
* It would be better if the authors can provide references for the temporal logic formula in the background section.
* There is a number of works for learning logical rules (e.g., Neural-lp, DRUM). It would be better to claim that the key difference between temporal logical rules and ordinary logical rules, and how temporal logical rules generalize them.
* The following paper is also relevant, please discuss this paper in the related work section.

Wang, Po-Wei, et al. "Differentiable learning of numerical rules in knowledge graphs." International Conference on Learning Representations. 2019.



**Summary Of The Paper:**

This paper proposed a model for learning temporal logical rules for the temporal point process. Specifically, the key idea is to formulate the rule learning as a restricted master problem and gradually optimize the objective function, which can significantly lower down the search space.
The authors conduct extensive experiments on both synthetic data and real data.


**Summary Of The Review:**

In summary, it seems that this is a novel and interesting paper for learning temporal logical rules with solid experiment results. I am familiar with logical rule learning but not the temporal point process, so I am not sure about my evaluation of the temporal process part. But in general, learning temporal logical rules is very interesting.

---

> ### Author Response · Authors · 2021-11-22
> **In response to reviewer GzHz**
>
> We thank the reviewer for the overall positive feedback.
>
> **As for the background regarding temporal logic rules**
>
> We have added the temporal logic formula references to the background section.
>
> A well written and comprehensive review regarding this is
>
> Goranko, Valentin and Antje Rumberg, "Temporal Logic", The Stanford Encyclopedia of Philosophy (Fall 2021 Edition), Edward N. Zalta (ed.), URL = <https://plato.stanford.edu/archives/fall2021/entries/logic-temporal/>.
>
> Put it in a simple way, the key difference between temporal logical rules and ordinary logical rules is that temporal logic rules contain extra temporal-related descriptions to define a logic formula. Formally, temporal logic rules have two kinds of operators: **logical operators** and **modal operators**. Logical operators are usual truth-functional operators (¬,∨,∧,→) just as in ordinary logic rules. The modal operators are used in linear temporal logic such as A Until B or in interval-based temporal logic rules such as A Before B, A Overlapping B, and so on.
>
> In this paper, we are using the **“interval-based temporal logic”** language originally proposed by Allen,
>
> Allen, James F. "Maintaining knowledge about temporal intervals." Communications of the ACM 26, no. 11 (1983): 832-843
>
> which is one of the widely used temporal logic languages.
>
> The reasons are below: This type of temporal logic rules are often suitable for reasoning about **events with duration**, which are better modeled if the underlying temporal ontology uses **time intervals**. It fits well to the modeling framework of **temporal point processes** for event sequences, where the time intervals of events are explicitly modeled.
>
> Another widely used temporal logic is called “linear temporal logic” (LTL) which uses tense operators in the rules. LTL is widely used in control, robotics, and reinforcement learning and can better model the policy related to states, actions, subgoals with temporal orders.
>
> **As for the references regarding logic rule learning**
>
> We have added the suggested references Neural LP, DRUM and
>
> “Wang, Po-Wei, et al. "Differentiable learning of numerical rules in knowledge graphs."
> International Conference on Learning Representations. 2019.”
>
> to our related work. To summarize, this branch of works focuses on learning the “ordinary logic rules” using an end-to-end differentiable approach. Both parameters and structures are jointly learned.
>
> “Wang, Po-Wei, et al. Differentiable learning of numerical rules in knowledge graphs." further extends Neural LP to efficiently learn numerical rules, which adds the expressiveness of the rule sets.
>
> As for how to design an end-to-end differentiable approach to learning temporal logic rules, it is still an open problem. Temporal logic rules can not be naturally evaluated via matrix-vector multiplications, due to the temporal structure. We may explore this direction in our future work.

---

> > ### Comment · Reviewer_GzHz · 2021-11-26
> > **Response to the authors' response**
> >
> > Thank you for your responses. All of my concerns are well addressed. I keep my rating as 6.

---

### Official Review · Reviewer_BWwC · 2021-11-04

**Correctness:** 3
**Technical Novelty And Significance:** 3
**Empirical Novelty And Significance:** 4
**Recommendation:** 8
**Confidence:** 3

**Main Review:**

I find the paper quite interesting. Admittedly, the proposed optimization problem is a very difficult one due to the high dimensionality of the parameter space and it is almost hopeless to find the global optimum. The proposed TELLER algorithm is not perfect, but the authors explain the idea very well and the algorithm makes heuristic sense. The problem this paper aims at solving is an important one and I can see that it will make a difference in many applications. I especially like the fact that domain experts were brought into the real data analysis and the results look interesting. The only concern I have is about the stability of the algorithm, which may be of critical importance in practice. Specifically,

1. In equation (6), how should the complexity $c_{f} $ be determined? I think they determine the sparsity of the solution $w$, hence is important. Simply saying that "$c_f$ can be the rule length" is a bit arbitrary. Wouldn't it make more sense if a tuning parameter, say $\lambda$, is introduced to balance the negative likelihood and complexity penalty? In such case, the Master Probe would become minimizing $-\ell(w,b_0,b_1)+\lambda\sum_{f\in\bar{\mathcal{F}}}c_fw_f$ and $\lambda$ can be chosen by some sort of cross-validation.

2. How much impact does the initial subset $\mathcal{F}_0$ have on the final solution? Can some simulations be conducted to show the impact of $\mathcal{F}_0$？

3. In synthetic data experiments, the number of true logic rules is at most 4. What would happen if this number continues to grow? In this case, the search space for the optimum will be much larger and how well would the algorithm work?


**Summary Of The Paper:**

This paper builds upon the work of [1] and proposes an algorithm to simultaneously estimate the logic rules and the model parameters.  The estimation problem is formulated as the contained optimization problem (6), with both continuous and discrete parameters to optimize over. The proposed TELLER algorithm divided the original problem into a sequence of Restricted master problems and gradually expand the search space for a global optimum.  Subjective constraints and expert opinions can be incorporated into the search algorithm as well.


[1] Shuang Li, Lu Wang, Ruizhi Zhang, Xiaofu Chang, Xuqin Liu, Yao Xie, Yuan Qi, and Le Song. Temporal logic
point processes. In International Conference on Machine Learning, pp. 5990–6000. PMLR, 2020.

**Summary Of The Review:**

 The problem this paper aims at solving is an important one and I can see that it will make a difference in many applications.

---

> ### Author Response · Authors · 2021-11-22
> **In response to reviewer BWwC**
>
> Thank you for the careful reading, nice comments, and constructive suggestions. We are grateful that you appreciated our work, including the real-world impact, novelty, and solid experiments. In the following response, we will provide detailed explanations to address each of your concerns point-by-point.
>
> **As for the tuning parameter and the complexity term**
>
> It is a great idea to introduce a tuning parameter $\lambda$ to balance the negative likelihood and the complexity penalty. $\lambda$ can be merged to the coefficients $c_f$ of the affine. As you suggested, any affine function of $w_f$, such as $\lambda \sum_{f\in \bar{\mathcal{F}}}c_fw_f$with $c_f\geq 0$, $\lambda \geq 0$ should work here and will not change the convexity of problem. Choosing $c_f$ as the rule length in the paper is to reflect our model assumptions that the true rules will be both sparse and relatively short. We followed your suggestions and tuned $\lambda=( 0.1, 0.5, 1, 5, 10, 100)$ in our experiments. For example, on synthetic dataset 1, we observed that a smaller $\lambda=0.1$ performed best.
>
> We have incorporated your suggestions and revised the descriptions after Equation (7) to clarify this.
>
>
> **As for the initialization of the subset $\mathcal{F}_0$**
>
> We empirically checked this when we were doing the synthetic experiments. For example, we tried randomly generating a short rule with only one body condition to initialize $\mathcal{F}_0$. The choice of $\mathcal{F}_0$ will not influence the final solution that much but will impact the run time.
>
> A rule of thumb we discovered is to start with an “empty” $\mathcal{F}_0$, which refers to there being only a base term in the intensity function, i.e., intensity = exp(b). The initial estimate of b will be learned by MLE (i.e., restricted master problem). Then we can start adding a new candidate rule by the subproblem.
>
> In real applications, we can imagine TELLER can leverage domain knowledge to initialize $\mathcal{F}_0$. There is no need to learn known rules from scratch, although we didn't try this initialization in our experiments.
>
> We have also added the above discussions to section 3.4.
>
> **As for the performance under longer true rules**
>
>
> For the accuracy: As demonstrated in our synthetic experiments, the existence of long logic rules will indeed make the problem more challenging and require “more data” and “stronger signal” (i.e., higher rule weights) to have an accurate recovery result.
>
> For the efficiency: For TELLER, longer true rules will increase the search space but the bottleneck in computation is more from the number of predicates. We can leverage the "heredity principle" to generate long rules from existing important short rules (depth-first rule search) to scale with the rule length. Expert’s preference in rule templates, if accessible,  can be utilized to further squeeze the search space.
>
> Although bounding the maximal rule length in the algorithm might sacrifice TELLER’s accuracy if this assumption violates true scenarios, this will make the algorithm more computationally feasible when the number of predicates is big.

---

> > ### Comment · Reviewer_BWwC · 2021-11-24
> > **Response to response**
> >
> > All my questions have been answered satisfactorily. While I agree with reviewer AiEu that the current work lacks rigorous theoretical justification, I understand the difficulties in establishing the theoretical bounds of the proposed optimization problem. From an application point of view, I do think the algorithm is innovative, the problem under consideration is important and may potentially have big impacts. Therefore, I would like to keep an open mind on the theoretical flaws and adjust my rating to 8.

---

### Official Review · Reviewer_sdkz · 2021-11-08

**Correctness:** 4
**Technical Novelty And Significance:** 3
**Empirical Novelty And Significance:** 3
**Recommendation:** 6
**Confidence:** 2

**Main Review:**

Intepretable models are increasingly relevant as ML models are applied to more and more domains where safety and fairness are important -- characteristics that are hard to judge for traditional black box models. The method presented is novel and gets good improvements on existing state of the art methods on synthetic and real data.

**Summary Of The Paper:**

The paper uses a method inspired by column generation to continuously add temporal logic rules to their model to maximize the likelihood of the data. The goal is to generate a model that is highly interpretable. They compare against state of the art baselines and get good improvements.

**Summary Of The Review:**

Novel method which gets improvements on SOTA

---

> ### Author Response · Authors · 2021-11-22
> **In response to reviewer sdkz**
>
> We thank the reviewer for appreciating the motivation and acknowledging the novelty of our method.

---

### Decision · Program_Chairs · 2022-01-20

**Decision:**

Accept (Poster)

**Comment:**

This paper has been independently evaluated by four expert reviewers. After discussion with authors, three of them set their recommendations at marginal acceptance, one at straight accept. Perhaps the key criticism involved limited rigor of theoretical justification for the proposed method, but it appears to be applicable in practice as the empirical results suggest. All things considered, I am leaning towards recommending that this paper is accepted for ICLR 2022.